# Manipulating solvent fluidic dynamics for large-area perovskite film-formation and white light-emitting diodes

Guangyi Shi[1,4], Zongming Huang[1,4], Ran Qiao[2,4], Wenjing Chen[1], Zhijian Li[1], Yaping Li[3], Kai Mu[2], Ting Si[2] & Zhengguo Xiao ®[1] ✉

Presynthesized perovskite quantum dots are very promising for making films with different compositions, as they decouple crystallization and film-formation processes. However, fabricating large-area uniform films using perovskite quantum dots is still very challenging due to the complex fluidic dynamics of the solvents. Here, we report a robust film-formation approach using an environmental-friendly binary-solvent strategy. Nonbenzene solvents, n-octane and n-hexane, are mixed to manipulate the fluidic and evaporation dynamics of the perovskite quantum dot inks, resulting in balanced Marangoni flow, enhanced ink spreadability, and uniform solute-redistribution. We can therefore blade-coat large-area uniform perovskite films with different compositions using the same fabrication parameters. White and red perovskite light-emitting diodes incorporating blade-coated films exhibit a decent external quantum efficiency of 10.6% and 15.3% (0.04 cm²), and show a uniform emission up to 28 cm². This work represents a significant step toward the application of perovskite light-emitting diodes in flat panel solid-state lighting.

Metal halide perovskites (MHPs) have been demonstrated to be promising direct band gap semiconducting materials with remarkable optoelectronic properties[1–10]. One of the most attractive properties of MHPs is their tunable optoelectronic properties via composition engineering[11–14]. For example, their band gap can be tuned from 1.19 eV with a composition of formamidinium tin iodide ($FASnI_3$) to 3.06 eV with a composition of cesium lead chloride ($CsPbCl_3$)[11,15]. This is particularly important for light-emitting diodes (LEDs) with tunable emission wavelengths[16,17], as well as tandem perovskite solar cells. However, MHPs with different compositions are significantly different in solubility, crystal nucleation kinetics, and trap characteristics[18–21]. For example, for organic-inorganic hybrid perovskites, such as formamidinium lead iodide ($FAPbI_3$), the inorganic component lead iodide ($PbI_2$) first precipitates out because of its much lower solubility

(1.8 M) than formamidinium iodide (FAI) (9.2 M) in dimethylformamide at room temperature. In contrast, for the all-inorganic perovskite cesium lead bromide ($CsPbBr_3$), lead bromide ($PbBr_2$) has a relatively higher solubility (0.93 M) than cesium bromide (CsBr) (0.025 M), and the perovskite phase directly precipitates out from the solution phase[22]. In addition, the perovskite crystallization and film-formation processes occur simultaneously in solution-processed perovskite films[23,24]. The complex and diverse film-formation process of perovskites with different compositions often results in poorly controlled film morphology and a high density of defects, especially in large-area films made by mass fabrication techniques[25–27]. Moreover, separately optimizing MHP films with different compositions will consume much additional effort and cost, which is unfavorable for the commercialization of perovskite devices.

[1]Department of Physics, CAS Key Laboratory of Strongly-Coupled Quantum Matter Physics, University of Science and Technology of China, Hefei, Anhui 230026, China. [2]Department of Modern Mechanics, University of Science and Technology of China, Hefei, Anhui 230026, China. [3]Center for Micro- and Nanoscale Research and Fabrication, University of Science and Technology of China, Hefei, Anhui 230026, China. [4]These authors contributed equally: Guangyi Shi, Zongming Huang, Ran Qiao. ✉e-mail: zhengguo@ustc.edu.cn

Using premade colloidal perovskite nanocrystals (PNCs) or perovskite quantum dots (PQDs) to prepare films can decouple perovskite crystallization from the film-formation process[28]. The fluidic properties such as surface tension, viscosity, and evaporation speed of the solvents determine their drying dynamics and solute redistribution[29-31]. Nonuniform surface tension due to spatial temperature gradients causes unmanageable solvent flow, such as Marangoni flow and capillary flow[32-34]. It is still very challenging to finely control the fluidic properties of PQD inks, which is key to the formation of uniform perovskite films.

In this work, we propose a robust environmentally friendly binary-solvent strategy for large-area PQD films with different compositions using the same fabrication parameters. Environmentally friendly n-octane and n-hexane are selected for the PQD inks due to their different saturated vapor pressures and fluidic properties. Compared with a single solvent, the solvent flows of the mixed solvent can be finely manipulated by changing the mixing ratio of the binary solvents. The blade-coated large-area films (54 cm$^2$) exhibit good uniformity in morphology and optoelectronic properties. The external quantum efficiency (EQE) of cesium lead iodide (CsPbI$_3$)-based red perovskite LEDs (PeLEDs) reaches 15.3% (0.04 cm$^2$). Large-area PeLEDs with a device area of 28 cm$^2$ also show very uniform emission. To exploit the full advantages of our binary-solvent approach of making uniform perovskite films, we further demonstrate a high-performance white PeLED (WPeLED) with an EQE over 10.6% (0.04 cm$^2$) by coupling a sky-blue PeLED (491 nm) with a layer of red PQDs (650 nm) as a down-converter. More importantly, we also show a large-area bright WPeLED (28 cm$^2$) with uniform emission, representing a significant step toward the application of PeLEDs in next-generation solid-state lighting.

## Results and discussion
### Mechanisms of manipulating the fluidic properties

Binary or ternary solvents have advantages of tunable viscosity and surface tension gradient to control the solvent flow of the wet film for uniform solute deposition. Compared to conventional QDs, PQDs are more ionic in nature and highly sensitive to polar solvents[35]. The selection principle of mixed solvents should include the following considerations. Firstly, the mixed solvents should well mediate with ligands to form a stable ink. Secondly, the mixed solvents should have a suitable boiling point to avoid forming nonuniform film caused by fast solvent evaporation, and also avoid uncontrolled deposition patterns (such as coffee ring) resulting from a slow drying process. Thirdly, the solvents should have relatively low viscosity and surface tension to ensure good spreadability for coating uniform thin films. Fourthly, the solvent used for mixing should be green and low toxicity to minimize the environmental and human health impact of coating processes.

Therefore, nonpolar solvents with saturated hydrocarbons are preferred to disperse PQDs. Meanwhile, they are halogen-free and more environmentally friendly than benzene solvents. Nonpolar solvents with long-chain alkanes or cycloalkanes, such as n-tridecane and decalin, have a relatively high boiling point, viscosity, and surface tension and are therefore not favorable for large-area coating processes. With the above considerations, solvents based on short-chain alkanes, namely, n-octane and n-hexane, were selected to disperse PQDs for large-area blade coating in this work. As mentioned above, the solvent flow, including Marangoni and capillary flow, affects the distribution/deposition of the PQDs and the film morphology.

For a single solvent system (Fig. 1a), such as n-hexane or n-octane, preferential solvent evaporation results in an inward thermal gradient from the apex to the edge of the droplets because the thickness of the droplet is nonuniform and the thermal conductivity of the substrate is generally much larger than these short-chain alkane solvents[36]. This inward temperature gradient causes a surface tension gradient and thus an inward Marangoni flow. The strength of the Marangoni flow

can be characterized by the Marangoni number, and the Marangoni number of the n-octane droplet reaches over 22,900 (see "Methods" section). This suggests that the thermal gradient-induced inward Marangoni flow is very strong and enhances solvent contraction. Therefore, the solute is transported by the inward Marangoni flow to the center. This center-focused deposition is also caused by the slow drying process of n-octane and the movable contact line, giving solutes enough time to migrate with the solvent instead of depositing on the substrate.

To suppress the inward Marangoni flow induced by the temperature gradient, we intentionally blend n-hexane with higher volatility and lower surface tension than n-octane (Fig. 1b). The boiling point, surface tension, and viscosity of the binary solvents all decrease as the volume percent (vol%) of n-hexane increases (Supplementary Fig. 1). For the binary solvent with proper n-hexane, the concentration of n-hexane at the edge of the ink droplets is lower than that at the center because of the faster solvent evaporation edge and the higher volatility of n-hexane. Therefore, surface tension gradients arising from spatial variations in n-hexane concentration cause an outward Marangoni flow. Such a surface tension gradient is larger than that induced by the temperature gradient[32]. Therefore, the n-hexane concentration-caused outward Marangoni flow plays a leading role and results in solvent spreading and uniform solute redistribution.

For the binary inks with excess n-hexane (Fig. 1c), the overstrong outward Marangoni flow brings solute to the edge of the droplet and induces a thicker film at the edge. Moreover, the spreading process of the droplets is strongly coupled with the fast drying step due to the fast evaporation rate of n-hexane, and the droplet dries during the spreading stage. This leads to weakened droplet spreading and an uncontrollable solute redistribution process.

We simulated the solvent flow in the droplets using the COMSOL platform based on the finite element method (Supplementary Method 1). The computational domain is shown in Supplementary Fig. 2. Both the top view (Fig. 1d–f) and the side view (Supplementary Fig. 3) results confirm the shift from inward to outward Marangoni flow after mixing n-octane with n-hexane (Fig. 1d, e), as well as the overstrong outward Marangoni flow and weakened droplet spreading with excess n-hexane (Fig. 1f).

### Preparation and characterization of perovskite films

We prepared perovskite films with different A-sites and X-sites (FAPbI$_3$, CsPbI$_3$, and CsPbBr$_3$) to prove the robustness of the binary-solvent approach. The details of the PQD synthesis are shown in Supplementary Fig. 4 and the "Methods" section. All PQDs with different compositions could be well dispersed in both solvent systems (Supplementary Fig. 5), and the PQD inks maintained good performance after storage for 30 days due to the good ligand-mediated solvation of PQDs in our binary solvents (Supplementary Fig. 6). We drop cast the PQD inks on a 3 × 3 cm$^2$ glass substrate to examine the flow of the solvent and the film-forming process. For a single solvent of octane, most PQDs accumulate at the center of the droplet rather than spread on the substrate during the drying step (Supplementary Fig. 7). This center-focused pattern is also consistent with previous reports[37]. For binary solvents, we obtained uniform PQD films with low n-hexane concentrations of 20 vol% and 40 vol%, and the corresponding contact angles of the PQD solutions reached zero (Supplementary Fig. 8). Such Marangoni-enhanced spreading is also confirmed by the larger spreading area of the PQD films. The PQD inks with different compositions have similar contact angles and spreading areas (Supplementary Fig. 8), and the drop-cast PQD films have similar shapes accordingly. It is noted that our binary-solvent strategy is also applicable to traditional core-shell cadmium selenide/zinc sulfide (CdSe/ZnS) QDs. As shown in Supplementary Fig. 7d, although the composition, surface ligands, and structure of CdSe/ZnS QDs are completely different from those of PQDs, we can still obtain almost the same

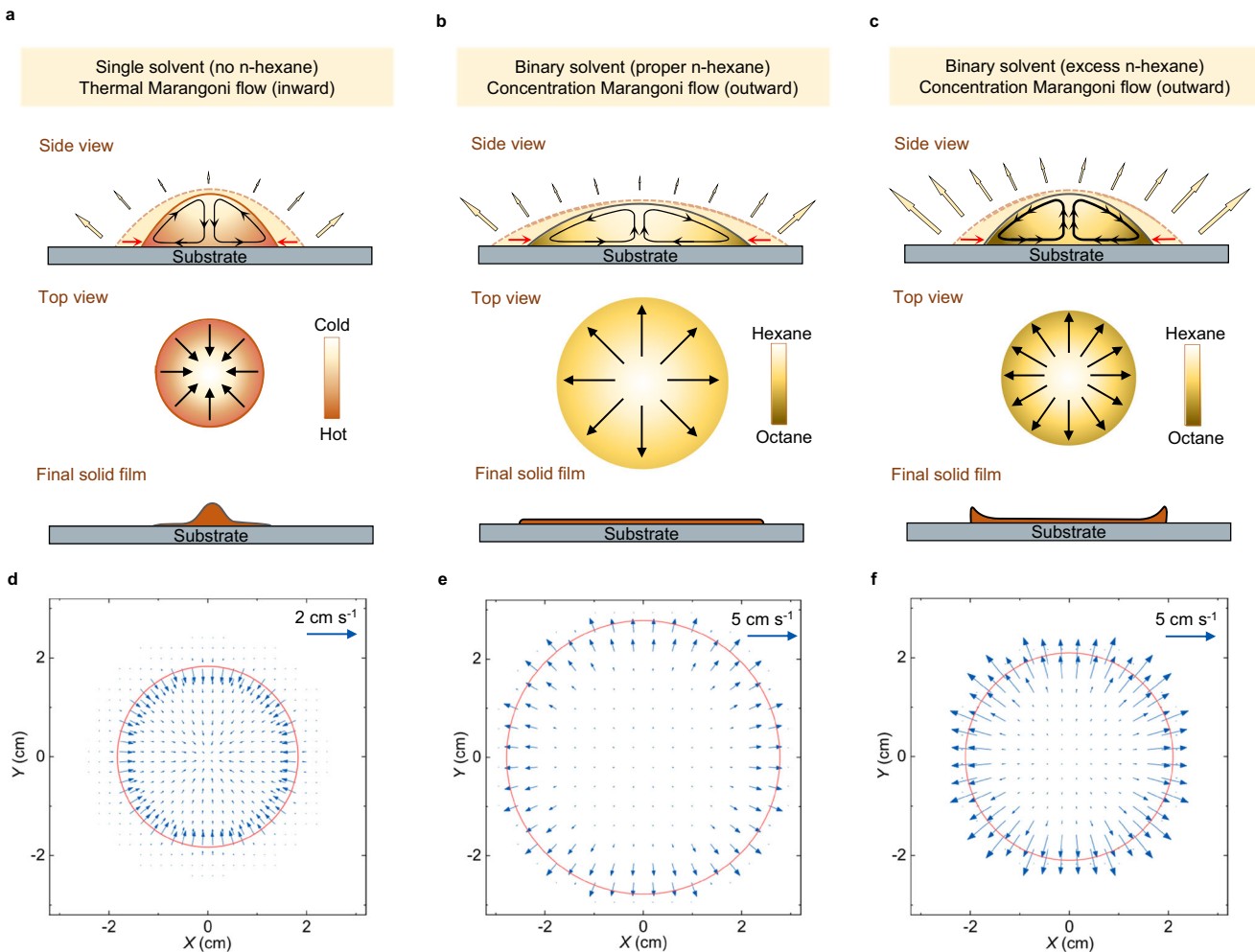

**Fig. 1 | Mechanism of manipulating fluid dynamics using binary-solvent system.** **a**–**c** Schematic illustration of solution droplet evaporation and final film patterns with zero (**a**), proper (**b**), and excess (**c**) ratios of n-hexane. The black arrows represent the flow pattern, and the red arrows represent the moving direction of the unpinned contact line. For the single solvents, inward flow driven by the thermal Marangoni effect induces droplet contraction. For binary solvents with proper n-hexane, outward flow driven by the concentration Marangoni effect enhances droplet spreading. For the binary solvents with excess n-hexane, the outward flow becomes stronger, but the droplet dries in advance during the spreading process due to the fast evaporation rate of excess n-hexane. **d**–**f** Simulation of the velocity vector of solvent flow when the droplets reach equilibrium contact angles with n-hexane ratios of 0 volume percent (vol%) (**d**), 20 vol% (**e**), and 80 vol% (**f**). The blue arrows represent the velocity vector, and the red circle represents the edge position.

film-formation behavior and deposition patterns. Such spreading phenomena in the binary-solvent system were not observed in the single solvent n-heptane (Supplementary Fig. 9). When the n-hexane concentration further increases to over 60 vol%, the strong outward Marangoni flow brings solute to the edge of the droplet, and the PQD ink dries during the spreading stage. This leads to an increase in the contact angle, a decrease in the spreading area, and thicker films at the edge (Supplementary Fig. 8).

To further demonstrate the robustness of our binary-solvent strategy, the surface morphology of the drop-casted PQD films with different compositions prepared by different solvents was examined by atomic force microscopy (AFM). As shown in Fig. 2, the surface roughness of all PQD films first decreases with increasing n-hexane concentration and reaches 1 nm when the concentration of n-hexane is 20 vol% and 40 vol%. When the n-hexane concentration is increased to larger than 60 vol%, the roughness of all PQD films increases, consistent with the analysis above. The PQDs are packed densely in all films, although the roughness varies with the n-hexane ratio (Supplementary Fig. 10). The PQD films made from 20 vol% and 40 vol% n-hexane also have better optoelectronic properties, such as longer

carrier lifetimes and higher photoluminescence quantum yield (PLQY) (Supplementary Fig. 11 and Supplementary Table 1).

It is noted that the binary-solvent approach has been adopted in preparing perovskite films or crystals[34,38,39]. The purpose of using binary solvents includes tuning the solvent-solute interaction strength[38], tuning the evaporation speed[34], tuning the nucleation and crystallization process of perovskites[39], etc. Our binary solvent is designed to manipulate the fluidic properties of the solvent to achieve uniform large-scale PQD deposition, which is essentially different. The droplet spreading using our binary-solvent system is also essentially different from the works modifying the surface energy of the substrate[39]. As shown in Supplementary Fig. 12, we can obtain almost the same uniform films on substrates with very different surface energies, which confirms that our binary-solvent strategy is a robust approach.

**Blade coating of large-area perovskite films**

We extend the above principle to the blade-coating process of PQDs using $CsPbI_3$ as an example. Depending on the coating speed and drying speed, the meniscus process can be divided into two regimes, namely the Landau-Levich regime and the evaporation regime. For the

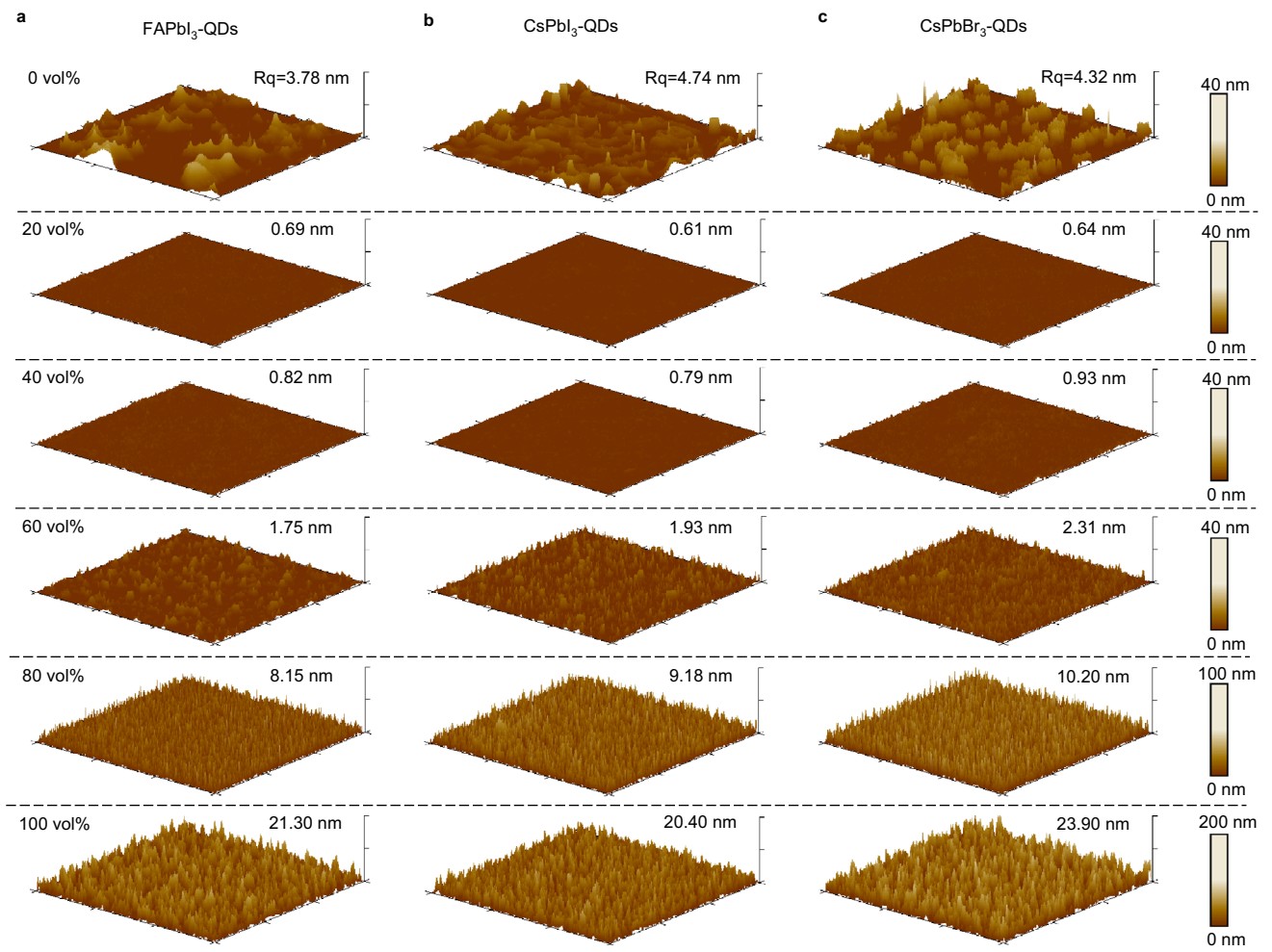

**Fig. 2 | Morphology study of perovskite quantum dot (PQD) films with different compositions. a–c** Atomic force microscopy (AFM) images of drop-cast FAPbI₃ films (**a**), CsPbI₃ films (**b**), and CsPbBr₃ films (**c**) with different n-hexane concentrations in the PQD inks. The concentration of n-hexane and roughness of the films are marked in the figures. The scan area of all images is 30 μm × 30 μm. Vol % is the volume percent. Rq is the root-mean-square surface roughness of PQD films.

single solvent system using n-octane with a high boiling point, the PQDs have enough time to migrate with the solvent rather than deposit on the substrate (Fig. 3a). Because of the inward Marangoni flow in the n-octane liquid films, PQDs are transported to the center and accumulate locally (Supplementary Fig. 13a). For the binary-solvent system with proper n-hexane, balanced Marangoni flow results in solvent spreading and uniform solute redistribution as mentioned above (Fig. 3b). Meanwhile, the PQDs are deposited on the substrate rather than migrate with the receding contact line due to the increased evaporation rates, resulting in uniform PQDs packing (Supplementary Fig. 13b). The coating process changes to the evaporation regime when there is excess n-hexane, due to the too fast drying speed of the binary solvent. In this case, the PQDs accumulate and form a solid film near the contact line. The over-strong Marangoni flow towards the contact line induces the PQD inks to detach from the receding side of the blader. The over-fast drying also causes most PQDs to pack at the beginning of blading. These problems result in a nonuniform and discontinuous film-formation process (Supplementary Fig. 13c).

Supplementary Fig. 14 shows the morphology and photoluminescence (PL) images of the blade-coated CsPbI₃ PQD films with different solvents. The thickness profiles, optical images, and AFM and scanning electron microscopy (SEM) images confirmed the homogeneous film-formation mechanism using the binary-solvent system. It is worth noting that the surface roughness was reduced

from 6.38 nm to 0.59 nm without using an N₂ knife, demonstrating the robustness of the binary-solvent system in the large-area blade-coating process. Large-area PeLEDs have potential applications in lighting[10,22].

To further demonstrate the scalability of our blade-coated PQD films using the binary solvent, a large-area CsPbI₃ film of 6 × 9 cm² was fabricated using inks with 20 vol% n-hexane without using an N₂ knife (Fig. 3c). We first measured the surface morphology of the large-area PQD film at four different positions (Fig. 3d). The roughness of blade-coated PQD films is almost the same over the whole substrate with a roughness of approximately 1.5 nm. The thickness profile also confirms the uniformity of the blade-coated PQD film at the macroscale (Fig. 3e). The large-area film shows very uniform PL emission (Fig. 3f). Large-area uniform CsPbBr₃, CdSe/ZnS, and FAPbI₃ QD films (6 × 9 cm²) can also be fabricated using the same parameters (Supplementary Fig. 15). We further cut the large-area films into small pieces 1 × 1 cm² in size to examine the uniformity of optoelectronic properties. The PLQY and average carrier lifetime ($\tau_{average}$) of the blade-coated thin film reach decent values of 45% and 42 ns, respectively, and are also very uniform over the whole film (Fig. 3g, h). As expected, the radiative and non-radiative recombination rates, obtained according to the formulas $1/\tau_{average} = k_{rad} + k_{nonrad}$ and $PLQY = k_{rad}/(k_{rad} + k_{nonrad})$, are also uniform with negligible fluctuations (Fig. 3i, j).

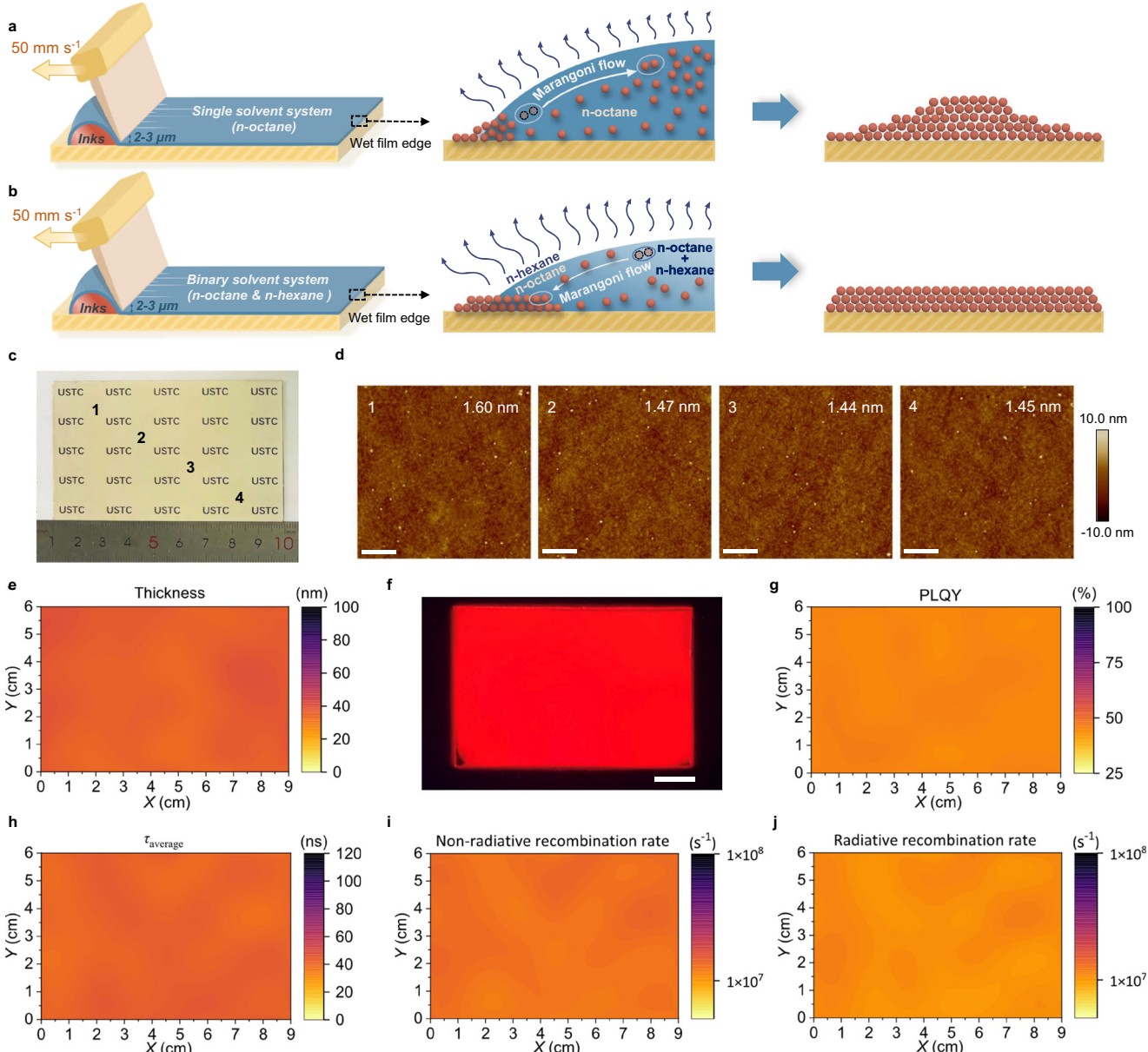

**Fig. 3 | Large-area perovskite quantum dot (PQD) films fabricated by blade coating. a, b** Schematics of the fabrication process of PQD films by using blade coating with a single solvent (**a**) and binary-solvent system (**b**). The red ball represents the PQD, the white arrow represents the direction of solvent flow, and the dark blue arrow represents the intensity of solvent evaporation. The concentration of CsPbI₃ quantum dot ink is 30 mg ml⁻¹. **c** Photo image of a large-area (6 × 9 cm²) perovskite film. **d** Atomic force microscopy (AFM) images of the large-area PQD film at different positions marked in (**c**). The roughness is marked in the images. The scale bars are 5 µm. **e** Thickness mapping of the large-area PQD film. **f** PL image of the large-area PQD film and the scale bar is 1.5 cm. **g–j** Photoluminescence quantum yield (PLQY) (**g**), average carrier lifetime ($\tau_{average}$) (**h**), and non-radiative (**i**) and radiative (**j**) recombination rate mapping of a large-area PQD film. The large-area films were divided into 1 × 1 cm² pieces, and thickness and optoelectronic characterizations were performed on these small pieces separately. The two-dimensional mappings are plotted by spatial interpolation.

## Efficient devices fabricated by blade coating

We fabricated PeLEDs using blade-coated CsPbI₃ PQD films with a structure of ITO/PEDOT:PSS/poly-TPD/CsPbI₃ PQDs/TPBi/LiF/Al (ITO: indium tin oxide); PEDOT:PSS: poly(3,4-ethylenedioxythiophene): poly(styrenesulfonate); poly-TPD: poly(N,N′-bis(4-butylphenyl)-N,N′-bis(phenyl)-benzidine; TPBi: 1,3,5-tris(1-phenyl-1H-benzimidazol-2-yl) benzene; LiF: lithium fluoride) (Fig. 4a). The cross-section SEM image and electroluminescence (EL) spectra of the device are shown in Fig. 4b, c, respectively. The energy diagram and angular intensity profiles are shown in Supplementary Fig. 16. Current density (*J*) and luminance (*L*) curves as a function of voltage (*V*) for CsPbI₃ devices are shown in Fig. 4d, e. The device with 20 vol% hexane features the lowest

turn-on voltage of 2.3 V at a luminance of 1 cd m⁻². Additionally, the peak EQE with 20 vol% hexane reaches 15.3%, much higher than that of the control device of 8.0%, due to the excellent film uniformity and optoelectronic properties (Fig. 4f).

We also demonstrated ultra-large PeLEDs with a device area of 4 × 7 cm². Notably, the hole-transport layers (stacked PEDOT:PSS and poly-TPD) are also fabricated by blade coating to be compatible with the fabrication of the emission layer (see "Methods" section). As shown in Supplementary Fig. 17, the blade-coated PEDOT:PSS and poly-TPD stacked layers are very flat with roughnesses of approximately 1.03 nm and 1.21 nm, respectively. The uniformity of the poly-TPD layer is also confirmed by its uniform PL emission (Fig. 4g). Under both low and

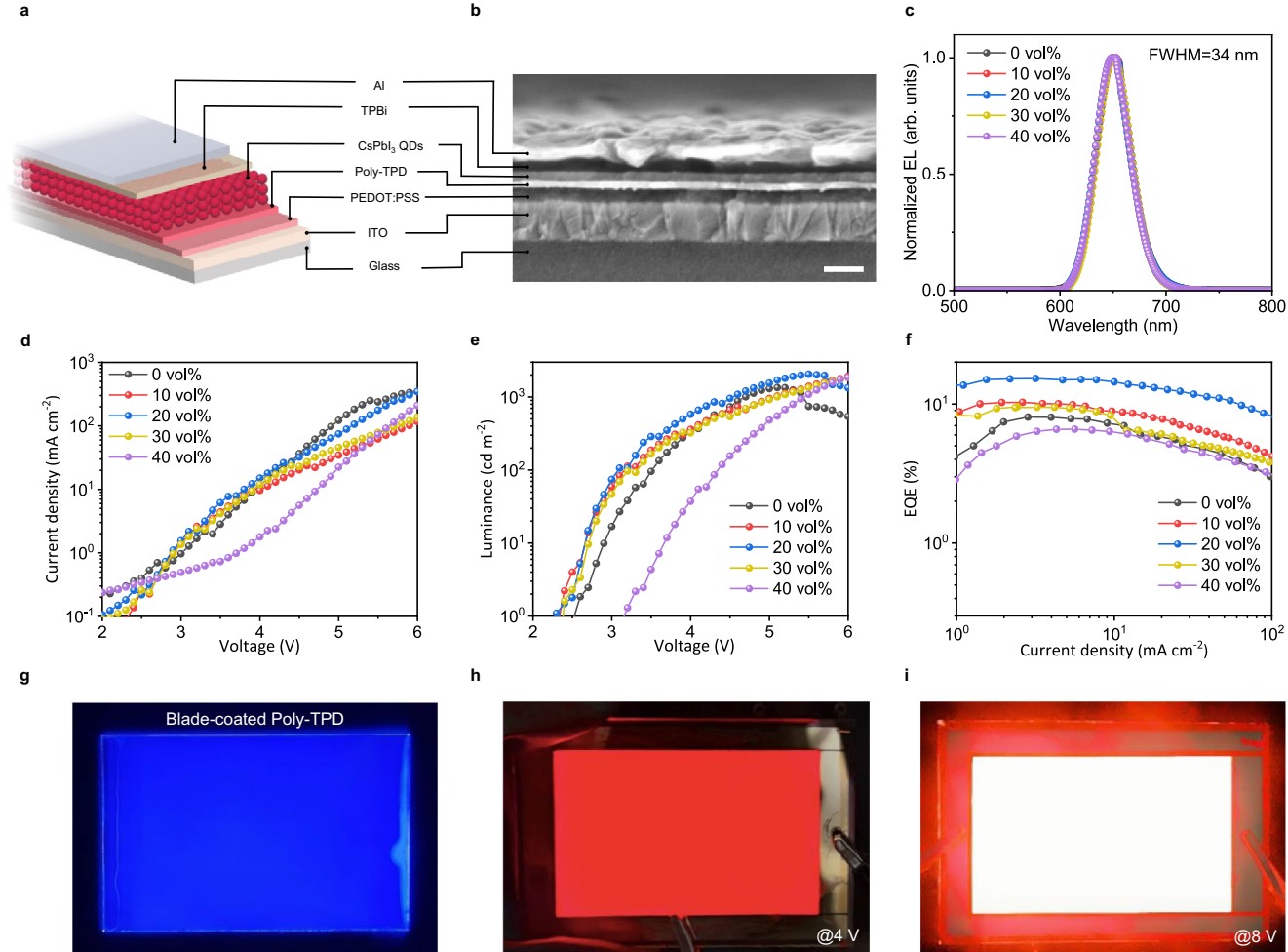

**Fig. 4 | Characterization of blade-coated perovskite light-emitting diodes (PeLEDs). a, b** Device structure (**a**), cross-sectional scanning electron microscopy (SEM) image (**b**) of PeLEDs. The scale bar is 100 nm. **c,** Electroluminescence (EL) spectra of blade-coated PeLEDs with different n-hexane concentrations. Vol% is the volume percent. FWHM is the full width at half maxima. **d–f** Current densities (**d**),

luminances (**e**), and external quantum efficiency (EQE) (**f**) of the PeLEDs fabricated with different n-hexane concentrations. **g** Photographs of the large-area ($6 \times 9 \, cm^2$) blade-coated polyN,N'-bis(4-butylphenyl)-N,N'-bis(phenyl)-benzidine (poly-TPD) layer. **h, i** Photograph of a large-area PeLED with a device area of $4 \times 7 \, cm^2$ under bias voltages of 4 V (**h**) and 8 V (**i**).

high driving voltages, the large-area PeLEDs show remarkably uniform light emission across the entire $28 \, cm^2$ device area (Fig. 4h, i). These excellent performances of the blade-coated large-area devices again demonstrate the possibility of scalable production of PeLEDs.

### Design and fabrication of white light-emitting diodes

Encouraged by the excellent photoluminescence of PQD films and the decent performance of PeLEDs fabricated by blade coating, we further combined their merits to construct large-area WPeLEDs. As shown in Fig. 5a, we designed an all-perovskite emission architecture consisting of sky-blue PeLEDs with a perovskite composition of $CsPb(Br_{0.84}Cl_{0.16})_3$ and a layer of red $CsPbI_3$ PQDs[16]. The $CsPbI_3$ PQDs were deposited on the glass side to convert part of the sky-blue light of 491 nm down to red light of 650 nm (Fig. 5b). The WPeLED color temperature can be well controlled from approximately 8000 K–3500 K by adjusting the thickness of the red PQD layer with different PQD concentrations (Fig. 5c).

The $J$-$V$-$L$ curves of the sky-blue PeLEDs and WPeLEDs are shown in Fig. 5d. The WPeLEDs feature a turn-on voltage of 2.9 V at a luminance of $1 \, cd \, m^{-2}$ and achieve a high value of approximately $4000 \, cd \, m^{-2}$ at a low bias of 4.3 V. The peak EQE of the WPeLEDs reaches 10.6%, slightly lower than its sky-blue counterpart of 11.8% (Fig. 5e). The WPeLEDs exhibit excellent spectral stability, demonstrating neglected phase segregation in the sky-blue PeLEDs and the red PQD downconverters

(Supplementary Fig. 18). The operating lifetime is still short and needs to be improved, similar to other PeLEDs in the field. To fully unleash the potential of our blade-coated WPeLEDs based on the all-perovskite emission architecture for solid-state lighting in large-scale industrial production, a large WPeLED was fabricated with a functional area up to $28 \, cm^2$ (Fig. 5f). The large-area WPeLED has bright and uniform white emission, showing potential in flat-panel lighting applications.

In summary, we have proposed a robust approach for manipulating the solvent fluidic dynamics of PQD inks by using a robust and environmentally friendly binary-solvent system. By tuning the ratio of n-hexane and n-octane with different boiling points and surface tensions, the solvent flows and the evaporation rate of the PQD liquid films can be well controlled, resulting in uniform deposition of PQDs with different compositions using exactly the same coating parameters. The thickness, roughness, and optoelectronic properties of the large-area blade-coated PQD films exhibit good uniformity. Consequently, the EQE of red PeLEDs reaches 15.3%. Moreover, we demonstrate high-performance WPeLEDs with EQEs over 10.6% by coupling a sky-blue PeLED with low ophthalmic toxicity with a red PQD downconverter layer and show a bright WPeLED with a large-area of $28 \, cm^2$ and uniform emission. This demonstration represents a significant step toward the realization of PeLEDs in next-generation healthy solid-state lighting.

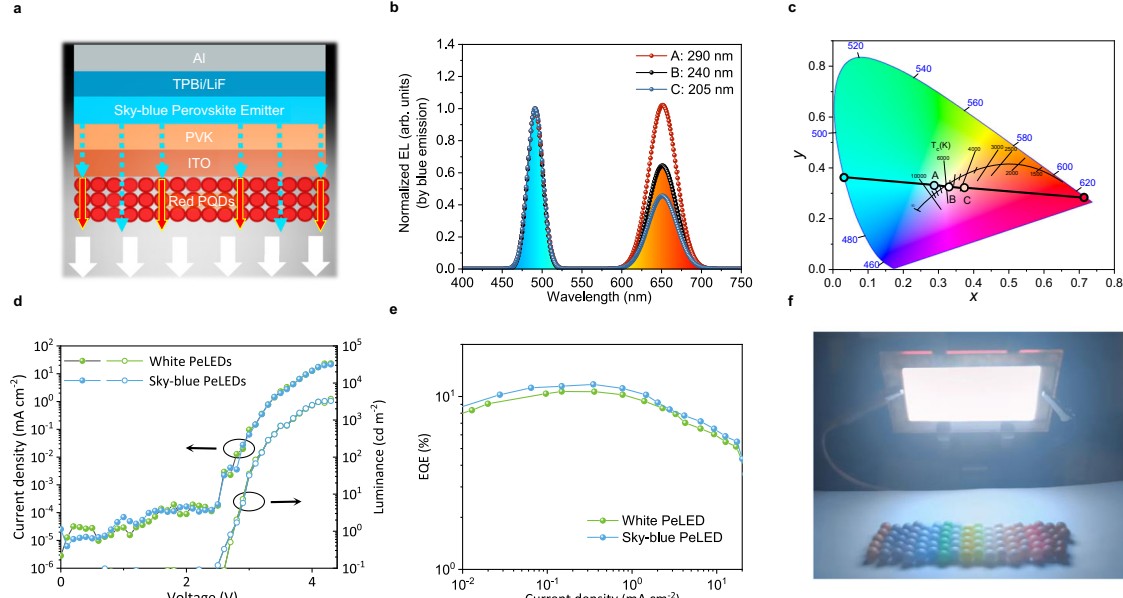

**Fig. 5 | Design and characterization of blade-coated large-area white perovskite light-emitting diodes (WPeLEDs). a** Device structure of WPeLEDs that couple sky-blue PeLEDs with a red perovskite quantum dot (PQD) downconverter layer to achieve white light emission. **b** Electroluminescence (EL) spectra of WPeLEDs normalized by the intensity of the sky-blue emission intensity. **c** CIE coordinates of WPeLEDs. Points A (0.29, 0.34), B (0.33, 0.33) and C (0.38, 0.32) represent 3 devices with various thicknesses of the red PQD layer in (**b**). **d, e** Current density-voltage-luminance curves (**d**) and external quantum efficiency (EQE) curves (**e**) of a WPeLED and its sky-blue counterpart. **f** Photo image of a large-area WPeLED with a device area of 4 × 7 cm².

## Methods

### Calculation of Marangoni number

The Marangoni number ($Ma$) is a dimensionless number that represents the transport rate due to Marangoni flow. It is determined by the equation $Ma \equiv -\beta(T_e - T_c) t_f / \mu R$, where $R$ is the contact-line radius, $T_e$ and $T_c$ are the surface temperatures at the edge and the top of the droplet respectively, $\beta$ is the temperature-dependent coefficient of surface tension, $t_f$ is the drying time, $\mu$ is the viscosity of the solution. For 3 μl n-octane inks, the contact-line radius $R$ is 5.2 mm, $\beta = -0.0935$ mN K$^{-1}$, and $\mu = 0.51 \times 10^{-3}$ Pa s.

### Perovskite quantum dots synthesis and purification

For CsPbI₃ QDs, 2 g cesium carbonate (Cs₂CO₃), 8 ml oleic acid (OA), and 100 ml 1-octadecene (ODE) were degassed in a flask at 100 °C for 30 min and then stirred at 120 °C for 1 h under N₂ to prepare the Cs-OA precursor. In another flask, 320 mg PbI₂, 320 mg zinc iodide (ZnI₂), 200 mg manganese acetate tetrahydrate ((CH₃COO)₂Mn·4H₂O), and 180 μL HI were mixed with 4 ml oleylamine (OAm)/OA mixture and 8 ml ODE followed by degassing at 100 °C for 30 min. Then, the solution was kept at 155 °C under N₂ until all solids were dissolved. Subsequently, 5 ml Cs-OA precursor was injected. Following a five-second interval, the solution was cooled by immersing the flask in an ice water bath. Subsequently, the solution underwent centrifugation at a speed of 6800 × $g$ to eliminate any remaining unreacted precursors. The resulting supernatant was then collected for further purification. For the purification process, the supernatant was precipitated with a 3-fold volume of methyl acetate. Then, the precipitate was collected by centrifuging the mixture at 6800 × $g$ for 5 min and dissolved in 5 ml n-hexane. This procedure was repeated once or twice. Finally, the supernatant was stored at 4 °C before use.

For CsPbBr₃ QDs, the Cs-OA precursor was prepared similarly to the CsPbI₃ QDs mentioned above. In a reaction flask, 700 mg PbBr₂ and 1400 mg zinc bromide (ZnBr₂) were mixed with 14 ml OAm, 14 ml OA, and 40 ml of ODE. The mixture was then degassed at 100 °C for 30 min. Subsequently, the solution was maintained under an N₂ atmosphere at 210 °C until all solid components were dissolved completely. Next, Cs-

OA precursor solution (11.0–11.8 ml) was injected into the reaction flask. After 10 s, the solution was cooled by immersing the flask in an ice water bath. Then, the solution underwent centrifugation at a speed of 6800 × $g$ to eliminate any remaining unreacted precursors. The resulting precipitant was collected for further purification. For the purification process, the resulting precipitate was dispersed in 7 ml of n-hexane and then 20 ml of methyl acetate was added. Then, the precipitate was collected by centrifuging the mixture at 6800 × $g$ for 5 min and dissolved in 7 ml n-hexane. This procedure was repeated once or twice. Finally, the supernatant was stored at 4 °C before use.

For the FAPbI₃ QDs, a flask was used to prepare the FA-OA precursor by mixing 521 mg FA-acetate and 10 ml OA. The mixture was degassed by heating at 120 °C for 60 min and then stirred under N₂ atmosphere at 80 °C for 60 min. In another flask, 175 mg PbI₂ was mixed with 1 ml OAm, 1 ml OA, and 10 mL ODE followed by degassing at 100 °C until all solid components were dissolved completely. Then, the solution was cooled to 80 °C, followed by a swift injection of 1.0 ml FA-OA precursor solution. Following a five-second interval, the solution was cooled by immersing the flask in an ice water bath. Then, the solution underwent centrifugation at a speed of 12,840 × $g$ to eliminate any remaining unreacted precursors. The resulting precipitant was collected for further purification. For the purification process, the resulting precipitate was dispersed in 7 ml of n-hexane and then add 7 ml ethyl acetate. Subsequently, the precipitate was collected by centrifuging the mixture at 12,840 × $g$ for 15 min and dissolved in 7 ml n-hexane. This procedure was repeated once or twice. Finally, the precipitate from the last purification procedure was dispersed in 2 ml n-hexane and then stored at 4 °C before use.

### Blade coating of perovskite and hole-transport films

The PQDs were dispersed in different solvents at a concentration of 30 mg mL⁻¹. We blade-coated the PQDs in air with a temperature and humidity of approximately 25 °C and 35%, respectively. The speed of the blader was set as 50 mm s⁻¹. The distance between the blader and substrate is approximately 2–3 μm. For blade coating of the PEDOT:PSS layer, the substrate was preheated to 40 °C, and the

PEDOT:PSS aqueous solution was diluted with methanol (PEDOT:PSS aqueous solution:methanol = 1:1, v/v). Then, the solution was blade-coated with a gap of 10 μm and a speed of 10 mm s$^{-1}$. For blade coating of the poly-TPD and poly(N-vinylcarbazole) (PVK) layers, the substrate was preheated to 50 °C. Then, the poly-TPD or PVK solution (6 mg ml$^{-1}$ in chlorobenzene) was blade-coated with a gap of 5 μm and a speed of 10 mm s$^{-1}$.

## Characterizations of perovskite inks and films

Contact angle measurement was carried out using SmartDrop software with a UNI-CAM (GITSOFTTECH) system. The surface tension was measured using KRÜSS BP100. The viscosity was measured using an Ubbelohde viscometer (Fungilab). We measured the thickness of the PQD films with the Dektak XT profiler. The AFM measurements were performed by Asylum Research MFP-3D-SA, operating under the tapping mode. The SEM images were collected in a GeminiSEM 500 with an accelerating voltage of 3 kV. Samples for transmission electron microscope (TEM) imaging were taken on a Hitachi HT7700 Exalens instrument with a 300 kV acceleration voltage. The PLQY of the PQD films was recorded using a Horiba Fluorolog-3 system with a Petite Integrating Sphere at an excitation wavelength of 405 nm. The TRPL was recorded using a Horiba time-correlated single-photon counting system at an excitation wavelength of 369 nm.

## Light-emitting diodes fabrication and characterization

For red PeLEDs, the PEDOT:PSS solution was spun or blade-coated onto the surface of the cleaned ITO substrate and baked at 140 °C for 20 min under ambient conditions. The poly-TPD, 6 mg ml$^{-1}$ in chlorobenzene, was spin-coated (2000 rpm) or blade-coated on the PEDOT:PSS layer and annealed at 150 °C on a hot plate for 20 min. Before coating PQDs, we conducted an $O_2$ plasma treatment on poly-TPD for 6 s to improve its wettability. We blade-coated the $CsPbI_3$ QDs in the air at room temperature with a humidity of 35%. The speed of the blader was set as 50 mm s$^{-1}$. The gap of the blader and substrate was set as 2–3 μm. After coating, the PQD films annealed at 60 °C for 10 min under ambient conditions. Finally, 40 nm TPBi, 1.2 nm LiF, and 100 nm Al were sequentially thermally evaporated under a high vacuum.

For white PeLEDs, the PVK, 3 mg ml$^{-1}$ in chlorobenzene, was spin-coated (2000 rpm) or blade-coated on the substrate and annealed at 120 °C for 20 min. Then we conducted an $O_2$ plasma treatment on PVK for 6 s to improve its wettability. Then, the sky-blue perovskite precursor solution was blade-coated on top of the PVK film at a speed of 50 mm s$^{-1}$. During the blade-coating process, the humidity and the substrate temperature were set at 30% and 50 °C, respectively. An $N_2$ knife was conducted to the blade-coated film for 60 s to remove excess solvents. The gap of the blader and substrate was set as 2–3 μm. After blade coating, the perovskite films were transferred to a nitrogen-purged glovebox without further annealing process. Then, 40 nm TPBi, 1.2 nm LiF, and 100 nm Al were sequentially thermally evaporated under a high vacuum. Finally, the red PQD inks (A: 180 mg ml$^{-1}$, B: 140 mg ml$^{-1}$, and C: 120 mg ml$^{-1}$) were blade-coated on the bottom of the substrate.

The current versus voltage characteristic of PeLEDs was measured using a Keithley 2400 sourcemeter. The luminescence was measured using a calibrated Si photodiode (FDS-100-CAL, Thorlabs) and a picoammeter (4140B, Agilent). The EL spectrum was collected using a calibrated fiber optic spectrophotometer (UVN-SR, StellarNet Inc.). All of the measurements were conducted at room temperature without encapsulation in an $N_2$ glovebox.

## Data availability

The data that support the findings of this study are provided in the main text and the Supplementary Information. More data are available from the corresponding author upon request.

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

## Acknowledgements

We acknowledge support from the National Key Research and Development Program of China (2022YFA1204800), the National Natural Science Foundation of China (62175226, 62234004), and the University Synergy Innovation Program of Anhui Province (GXXT-2022-009). This work was partially carried out at the University of Science and Technology of China Center for Micro and Nanoscale Research and Fabrication. We thank Jun Ma from the USTC Center for Micro and Nanoscale Research and Fabrication for the ellipsometry measurement. This work was partially carried out at the Instruments Center for Physical Science, University of Science and Technology of China.

## Author contributions

Z.X. and G.S. conceived the idea. G.S. designed and performed most of the experiments. Z.H. performed the PLQY and TRPL measurements. R.Q., K.M., and T.S. simulated the velocity vector distribution in the droplet. W.C. and Z.L. performed the thickness, SEM, and TEM characterizations. Y.L. helped with the AFM measurements. Z.X. and G.S. wrote the manuscript, and all authors discussed the results and reviewed the manuscript. Z.X. supervised the project.

## Competing interests

The authors declare no competing interests.
