## [Peer Review File · Nature Communications]

Reviewers' comments:

Reviewer #1 (Remarks to the Author):

The authors report a binary solvent approach to attain uniform perovskite thin film via blade coating, and achieved respectable LED efficiencies of up to 15%. The results are surprising because the authors have used 2 rather similar solvents in their binary mixture (hexane and octane), and yet achieved a notable improvement in film quality. I think this paper could be published upon some revisions.

1) It is not clear what device areas correspond to the devices with high EQE. The device areas should be clearly indicated in the abstract as well as in the main text for the CsPbI₃ and white PeLEDs. My impression is that the EQEs are reported on smaller devices, rather than the ultra large 28cm² device.

2) Does heptane (7 carbon) solvent give similar performance compared to binary mixture of octane (8 carbon) and hexane (6 carbon)?

3) Does the substrate matter for the binary solvent blade coating? I noted that the perovskite would have to be coated on different surfaces depending on their application in PeLEDs or as a photoluminescent film layer on glass.

4) The EL spectra for the CsPbI₃ should be shown in the main text figure for reference on emission wavelength.

5) Reference should probably be made to earlier works on large area perovskite LED (Nature Photonics 14, 215–218 (2020))

Reviewer #3 (Remarks to the Author):

Shi et al. report a co-solvent method for fabricating uniform films of colloidal quantum dot (CQD) films of inorganic and hybrid perovskites on small and medium scale substrates (considered large for the perovskite community). Achieving uniform films is important, but it is not the most vexing challenge the

community is facing. Nevertheless, the method shows facile pathway to large area fabrication of uniform CQD devices, which is important for the field to mature.

The work is therefore interesting and potentially impactful, but it is unclear whether it is a breakthrough and suitable for publication in a journal such as Nature Communications.

There are several issues:

(1) the title is a bit deceptive and I found myself disappointed when I read "universal film formation strategy" only to realize that the authors only demonstrate how co-solvents help achieve uniform films of perovskite quantum dots. These are pre-synthesized building blocks unlike the vast majority of halide perovskites that the community making emitters, detectors and harvesters will be using. This is therefore fairly easy and I was unimpressed with the achievement. I do not see it as a "breakthrough" or conceptual novelty on par with other film forming or crystallization studies using co-solvents, for instance 10.1038/s41467-021-27740-4 or 10.1002/adma.202109862, which used similar formulation concepts within traditional perovskites to achieve controlled wetting or dewetting. I note that these studies were not referenced. These studies demonstrated universality and additionally addressed phase transformation and/or material saving, ink stability and/or process sustainability, which this study does not.

- My recommendation is for this study to be published in a specialized journal where the readership is interested in optoelectronics/photonics and/or quantum dots and for the authors to clearly identify "perovskite quantum dots" in the title of the manuscript.

(2) The authors speak at length about the challenge of processing compositionally tuned perovskite materials. However, this issue is NOT addressed by THIS work. This issue has been previously addressed by creating CQDs of perovskites which can be suspended in solvents depending on the choice of ligand. This issue has therefore been resolved. The current study addresses uniform film coating.

"Here, we report a perovskite composition-independent film formation approach via an environment-friendly binary-solvent strategy. Presynthesized perovskite quantum dots (PQDs) are used to decouple the perovskite crystallization and film formation processes."

With the above statement, it seems that the authors are claiming to have invented the concept of colloidal nanocrystals and colloidal quantum dots! What they are stating is obvious to any practitioner in this field: crystalline materials are pre-synthesized, then ligand capped, suspended in a solvent and processing can be done in a manner that is (nearly) independent of the nanocrystal's internal structure and composition.

(3) The authors do not provide a satisfactory explanation for why the QD film roughness decreases with the optimal co-solvent ratio and once again increases with excess co-solvent. Beyond cartoon schematics, they are encouraged to model the growth behavior and validate this.

Reviewer #4 (Remarks to the Author):

This work reports a solvent engineering to deposit large-area and uniform perovskite quantum dots films of different compositions by blade coating. A binary-solvent strategy is developed which allows to achieve low surface energy, thereby enhancing its wettability, and eventually improves the film homogeneity. The authors carefully organized the manuscript and tried to sell a sound story. However, the entire work focuses too much on the engineering aspects with little scientific contribution in the field. In addition, the EQEs of the prepared small-area LED devices are rather decent as compared to the state-of-the-art perovskite QLED devices. As such, I would recommend this manuscript to be considered in a more specific journal based on, additionally, the following comments.

1. In the abstract, the authors highlight “Presynthesized perovskite quantum dots (PQDs) are used to decouple the perovskite crystallization and film formation processes”. This sentence is obviously meaningless, as we know that film formation of quantum dots from a dispersion solution doesn’t involve any nucleation and crystallization processes. The authors don’t necessarily relate the deposition of quantum dots film to the perovskite formation from precursor-based solutions.
2. Solvent engineering, including binary and ternary solvent strategies, has been well studied in spin-coating to deposit large-area perovskite LED devices. The principle behind the solvent engineering for blade-coating in the present work should be the same as previous studies on spin-coating, which is to reduce surface energy and increase wettability of the formulation.
3. Related works should be cited in the design of the white LED in combination with sky blue perovskite LED devices.

Reviewer #5 (Remarks to the Author):

This manuscript as noted by Shi and co-workers demonstrated a binary-solvent strategy (n-octane and n-hexane) for fabrication of large-area perovskite light-emitting diodes (PeLEDs). With this binary solvent (n-octane and n-hexane) recipe, a blade-coated uniform perovskite quantum dot (PQD) thin film forming process is obtained. As a result, a good external quantum efficiency (EQE) of 15.3% is achieved in PeLEDs with the blade-coated PQD thin film. Large-area white PeLEDs (WPeLEDs) with area of 28 cm² by coupling a sky-blue PeLED with a layer of red PQDs as a downconverter is further demonstrated, and an

EQE of 10.6% is realized. However, similar strategy has already been reported by the other groups (Advanced Materials, 2022, adma.202107798; Micromachines, 2022, 13, 983; Nano Research, 2021, 14, 4125–4131), and large-area PeLEDs have already been presented by the same group (Advanced Materials, 2022, adma.202108939; Nature Communications, 2021, 12, 147). This paper mainly focuses on optimizing the fabrication processes of the devices. Although the performance of PeLEDs is improved, the mechanisms are not so clear and further verifications are needed. Considering the limited novelty (fabrication processes optimization) of the paper and very little scientific contribution to field, I do not recommend its to be published in high impact journals like Nature Communications. The paper seems to be more suited for other specialized journals after addressing below issues, eg. Light: Science & Applications.

Questions:

1. In Figure S3., the authors claims that 0 vol% and 100 vol% is capillary spreading, 20 vol% and 40 vol% is the Marangoni enhanced spreading range, and the 60 vol% and 80 vol% is evaporation weakened spreading. However, if there is a fluid volatilization, there is a Marangoni flow effect. The detailed mechanism and why the PQDs (different types) could form uniform thin films, how is the PQDs packing during the processing are not been clearly investigated and understood.
2. What about the long-term storage stability of the n-octane and n-hexane-based PQD inks?
3. The packing, defects and electrical properties of the PQD thin film have not been carefully investigated.
4. More detail of PQD ink preparation and PeLED device fabrication should be further supplied to make this work more repeatable to the readers.

Response Letter

We thank all reviewers for their valuable comments and criticism, which have helped us improve the manuscript's quality significantly. In the past two months, we tried our best to perform additional experiments and simulations. We also tried our best to clarify the essential differences between our work and previous works. We have addressed the reviewer's comments point-by-point and modified the manuscript accordingly.

Reviewer #1:

Comments: The authors report a binary solvent approach to attain uniform perovskite thin film via blade coating, and achieved respectable LED efficiencies of up to 15%. The results are surprising because the authors have used 2 rather similar solvents in their binary mixture (hexane and octane), and yet achieved a notable improvement in film quality. I think this paper could be published upon some revisions.

Response: We appreciate the reviewer's acknowledgment of our work and the constructive comments that helped us improve the quality of our manuscript. We have tried our best to address the comments point-by-point as follows.

Comment 1: It is not clear what device areas correspond to the devices with high EQE. The device areas should be clearly indicated in the abstract as well as in the main text for the CsPbI₃ and white PeLEDs. My impression is that the EQEs are reported on smaller devices, rather than the ultra large 28 cm² device.

Response: We appreciate the reviewer's reminder and apologize for not marking the device area corresponding to the peak EQEs. The device area for the peak EQE is 0.04 cm². In our lab, we use the following setup to measure the device performance and assume a point light source (Fig. R1-1). The current-voltage curve of the PeLEDs is measured using a source-measure unit. A calibrated Si photodiode placed in the front direction of the PeLEDs is used to detect emitted light assuming PeLEDs as a point light source. The distance between the PeLEDs and the photodetector is 60 mm, much larger than the size of the PeLEDs (2 mm). The EL spectrum of the PeLEDs was measured by a spectrometer coupled to an optical fiber. The angular-dependent emission intensity and spectrum are determined by rotating the Si photodiode controlled by a step motor.

We apologize that we cannot measure the large-area PeLEDs (28 cm²) using this setup, although the emission is very uniform. We have marked the device areas in the abstract and the main text in the revised version.

Fig. R1-1 | Schematics of the PeLED measurement setup. The schematics are adapted from *Nat. Photon.* 2019, 13, 818.

Comment 2: Does heptane (7 carbon) solvent give similar performance compared to binary mixture of octane (8 carbon) and hexane (6 carbon)?

Response: This is a very clever point! We followed the suggestion and tested the solvent heptane. As shown in the following figure, although the surface tension, viscosity, and boiling point of heptane are between those of n-hexane and n-octane, it could not result in uniform films. As a single solvent, the film formation mechanisms using heptane are closer to those using hexane or octane. This further demonstrates the unique advantages of binary-solvent in controlling the fluid characteristics of PQD inks.

Fig. R1-2 Morphology of PQD films based on n-heptane and our binary-solvent (20 vol% n-hexane). The volume of PQD inks is 3 μ l, and the scale bar is 1 cm. **(We added this figure as Supplementary Fig. 9 in the revised manuscript)**

We modified the manuscript accordingly:

1. On page 6, line 19 of the revised manuscript, we added: “*Such spreading phenomena in the binary-solvent system were not observed in the single solvent n-heptane (Supplementary Fig. 9).*”

Comment 3: Does the substrate matter for the binary solvent blade coating? I noted that the perovskite would have to be coated on different surfaces depending on their application in PeLEDs or as a photoluminescent film layer on glass.

Response: We thank the reviewer for this comment. It is true that perovskite films are usually coated on different electron/hole transport layers to improve the performance of PeLEDs. The strong spread ability of the binary-solvent used in our work is not related to the surface energy and wettability of the substrate. There are two approaches to improve the spreading of solvents on a substrate (*Phys. Rev. Lett.* 2021, 127, 024502). One approach is to improve the wetting of the substrate by reducing its surface energy. The second approach is to manipulate the fluidic and evaporation dynamics of the solvent, which is the case in our work.

We followed your suggestion and tested four substrates with different surface energies: glass, PEDOT:PSS, poly(9-vinylcarbazole) (PVK), and poly-TPD. As shown in Fig. R1-3, the influence of the substrate on the coating process is very minor because of the high spreading ability of our binary-solvents, and we obtained almost identical deposition patterns on these different surfaces.

Fig. R1-3 | Contact angle of different substrates (top panel) and morphology of PQD (20 vol% n-hexane) films on the substrates. The volume of PQD inks is 3 μ l, and the scale bar is 1 cm. (We added this figure as Supplementary Fig. 12 in the revised manuscript)

We modified the manuscript accordingly:

1. On page 7, line 19 of the revised manuscript, we added: “As shown in Supplementary Fig. 12, we can obtain almost the same uniform films on substrates with very different surface energies, which demonstrates that the binary-solvent strategy is a robust approach.”

Comment 4: The EL spectra for the CsPbI₃ should be shown in the main text figure for reference on emission wavelength.

Response: We thank the reviewer very much for the suggestion. In the revised manuscript, we replaced the energy level diagram in Fig. 4c in our original submission with the EL spectra for the CsPbI₃ PeLEDs with different n-hexane concentrations.

Fig. 4c | EL spectra of blade-coated PeLEDs with different n-hexane concentrations.

Comment 5: Reference should probably be made to earlier works on large area perovskite LED (Nature Photonics 14, 215-218 (2020))

Response: We thank the reviewer for the reference and apologize for missing this important work in our original submission. Tan’s work is a milestone in the PeLED field. We have cited this paper in our revised version as ref. #10 on page 2, line 7 and page 8, line 14 of the revised manuscript.

Reviewer #3:

Comments: Shi et al. report a co-solvent method for fabricating uniform films of colloidal quantum dot (CQD) films of inorganic and hybrid perovskites on small and medium scale substrates (considered large for the perovskite community). Achieving uniform films is important, but it is not the most vexing challenge the community is facing. Nevertheless, the method shows facile pathway to large area fabrication of uniform CQD devices, which is important for the field to mature.

The work is therefore interesting and potentially impactful, but it is unclear whether it is a breakthrough and suitable for publication in a journal such as Nature Communications.

Response: We thank the reviewer for the time and acknowledgment of our work. At present, the EQE of PeLEDs is close to that of state-of-the-art OLEDs and QLEDs. The stability of the PeLEDs also improved significantly in the past two years. **According to our judgment, PeLEDs are more likely to be first applied in SSL for the following two reasons.** First, it is still very challenging to deposit patterned red-green-blue perovskites on the TFT backplane, while OLEDs are already very mature in display. SSL does not need patterning. Second, perovskites have much higher conductivity than QLEDs and OLEDs and are therefore more suitable for SSLs, which require higher luminance. **We believe that large-area white PeLEDs (WPeLEDs) are promising as a revolutionary technology for SSL. However, large-area WPeLEDs have not been reported due to the lack of device design and fabrication techniques.**

In this work, two carefully selected solvents (n-octane and n-hexane) with different boiling points and fluidic properties were used to disperse the PQDs. This enables us to manipulate the fluidic and drying properties of perovskite inks and thus achieve a uniform film-forming process. Based on this strategy, we blade-coated high-quality large-area perovskite films with different compositions using the same parameters. All perovskite films show great uniformity in thickness, roughness, and optoelectronic properties. Furthermore, we demonstrate WPeLEDs with an EQE over 10.6% by coupling sky-blue PeLEDs with a layer of red PQDs as a downconverter. **A large-area WPeLED (28 cm²) with uniform emission was demonstrated for the first time, which we believe is a significant breakthrough in the PeLED field.**

Comment 1: the title is a bit deceptive and I found myself disappointed when I read "universal film formation strategy" only to realize that the authors only demonstrate how co-solvents help achieve uniform films of perovskite quantum dots. These are pre-synthesized building blocks unlike the vast majority of halide perovskites that the community making emitters, detectors and harvesters will be using. This is therefore fairly easy and I was unimpressed with the achievement.

I do not see it as a "breakthrough" or conceptual novelty on par with other film forming or crystallization studies using co-solvents, for instance 10.1038/s41467-021-27740-4 or 10.1002/adma.202109862, which used similar formulation concepts within traditional perovskites to achieve controlled wetting or dewetting. I note that these studies were not referenced. These studies demonstrated universality and additionally addressed phase transformation and/or material saving, ink stability and/or process sustainability, which this study does not.

Response: We apologize for the inappropriate term "universal". We used this term because we could make high-quality perovskite films with different compositions using exactly the same fabrication parameters. We replaced "universal" with "**robust**" and modified our description in

the revised manuscript. We also changed our title to make it more accurate: “*Manipulating solvent fluidic dynamics for large-area perovskite film formation and white light-emitting diodes*”

We thank the reviewer for pointing out the references. **The motivation, solvent design, and film formation mechanism are essentially different between our work and previous works.** **1. Motivation and solvent design:** In the first reference, the authors aimed to make thick films for perovskite solar cells. A cosolvent (THF:DMF) was designed to tune the solvent-solute interaction strength and accelerate the evaporation speed. In the second reference, the authors aimed to make precise arrays of single crystals for photodetectors. A cosolvent (n-cyclohexyl-2-pyrrolidone and DMF) was designed to achieve early supersaturation, nucleation, and rapid growth of perovskite single crystals through differential volatilization of the cosolvent. In fact, their cosolvent is good for contraction rather than spreading of the droplets, which is also in line with the goal of making arrays. In contrast, we aimed to develop a large-area deposition method for perovskite thin films. Environmentally friendly n-octane and n-hexane with different saturated vapor pressures and fluidic properties were designed to manipulate the solvent flow and solute redistribution inside the wet film. **2. Film formation mechanism:** The first work enhanced solvent spreading by adjusting the surface energy and wettability (Fig. R3-1a). The second work did not mention the relationship between solvents and spreading. Unlike these common mechanisms, our strategy is to form an outward Marangoni flow inside the wet film (Fig. R3-1b). After finding a balance between the strength of the outward Marangoni flow and the speed of drying, we ultimately achieved uniform deposition of PQD thin films.

Fig. R3-1 | Two mechanisms of making solvent spread on a substrate. Schematic diagram of capillary-driven spreading (a) and Marangoni-driven spreading (b). (*Phys. Rev. Lett.* 2021, 127, 024502). The former highly depends on the surface energy of the substrate.

In the past two months, we performed additional experiments and further demonstrated the robustness of our binary-solvent strategy. Here, we listed the achievements in our work, which may not be clear in our original submission, as follows. **1. Environmentally friendly solvent system for PQDs:** In nonpolar solvents commonly used for dispersing PQDs, n-octane and n-hexane are halogen-free and more environmentally friendly than benzene solvents. The PQD inks maintained good performance after storage in air for one month (Fig. R3-2). **2. Robustness of our cosolvent system:** In the revised manuscript, we provide more evidence to demonstrate the robustness of our approach. First, we can obtain almost identical deposition patterns on substrates with different wetting properties: glass, PEDOT:PSS, PVK, and poly-TPD (Fig. R3-3). This is due to the specific wetting mechanism of our cosolvent mentioned above. Second, our binary-solvent strategy is also applicable to traditional core-shell CdSe/ZnS QDs. As shown in Fig. R3-4, although the composition, surface ligands, and structure of CdSe/ZnS are completely different from those of perovskite QDs, we can still obtain almost the same film formation behavior and deposition patterns. A large-area uniform CdSe/ZnS QD film of $6 \times 9 \text{ cm}^2$ is

fabricated using ink with 20 vol% n-hexane (Fig. R3-5). **3. Simplified large-area fabrication approach:** In the blade-coating approach, a N_2 knife, substrate heating, or vacuum is usually adopted to enhance the solvent evaporation speed, which is very difficult to scale up. Our approach does not need this assistance, which makes it much easier for large-area and high-throughput production. **4. First demonstration of large-area WPeLEDs:** This represents a significant step toward SSL using perovskites as emitters, and we think it is a milestone in the field.

Fig. R3-2 | Photographs of $FAPbI_3$, $CsPbI_3$, and $CsPbBr_3$ QDs under room light (left) and a UV lamp at 365 nm (right) after aging in air at 4 °C for 30 days. The concentration of PQDs was 30 mg/mL. (We added this figure as Supplementary Fig. 6 in the revised manuscript)

Fig. R3-3 | Contact angle of different substrates (top panel) and morphology of PQD films on substrates made using cosolvent with 20 vol% n-hexane. The volume of PQD inks is 3 μ L, and

the scale bar is 1 cm. (We added this figure as Supplementary Fig. 12 of the revised manuscript)

Supplementary Fig. 7 | Morphology of drop-cast FAPbI₃ (a), CsPbI₃ (b), CsPbBr₃ (c), and CdSe/ZnS (d) QD films with different n-hexane concentrations. The volume of PQD inks is 3 μl, and the scale bar is 1 cm. (We added the data of CsSe/ZnS QDs as Supplementary Fig. 7d of the revised manuscript)

Fig. R3-5 | Large-area QD films with different compositions fabricated by blade coating. a-c, Photo image (a), PL image (b), and PL spectra (c) of $6 \times 9 \text{ cm}^2$ blade-coated CsPbBr₃, CdSe/ZnS, and FAPbI₃ QD films. (We added this figure as Supplementary Fig. 14 of the revised manuscript)

We modified the manuscript accordingly:

1. We changed our title to: “Manipulating solvent fluidic dynamics for large-area perovskite film formation and white light-emitting diodes”
2. On page 7, line 13 of the revised manuscript, we cited the references and added the critical differences between our approach and previous approaches on cosolvent and modifying the surface energy: “It is noted that the binary-solvent approach has been adopted in preparing perovskite films or crystals³⁸⁻⁴⁰. The purpose of using a binary solvent includes tuning the solvent-solute interaction strength³⁸, tuning the evaporation speed³⁹, tuning the nucleation and crystallization process of perovskites⁴⁰, etc. Our binary solvent is designed to manipulate the fluidic properties of the solvent to achieve uniform large-scale PQD deposition, which is essentially different. The droplet spreading using our binary-solvent system is also essentially different from the works modifying the surface energy of the substrate⁴⁰. As shown in Supplementary Fig. 12, we can obtain almost the same uniform films on substrates with very different surface energies, which demonstrates that the binary-solvent strategy is a robust approach.”

Comment 2: The authors speak at length about the challenge of processing compositionally tuned perovskite materials. However, this issue is NOT addressed by THIS work. This issue has been previously addressed by creating CQDs of perovskites which can be suspended in solvents depending on the choice of ligand. This issue has therefore been resolved. The current study addresses uniform film coating.

"Here, we report a perovskite composition-independent film formation approach via an environment-friendly binary-solvent strategy. Presynthesized perovskite quantum dots (PQDs) are used to decouple the perovskite crystallization and film formation processes."

With the above statement, it seems that the authors are claiming to have invented the concept of colloidal nanocrystals and colloidal quantum dots! What they are stating is obvious to any practitioner in this field: crystalline materials are pre-synthesized, then ligand capped, suspended in a solvent and processing can be done in a manner that is (nearly) independent of the nanocrystal's internal structure and composition.

Response: We thank the reviewer very much for pointing out the misunderstanding statements, and we apologize for this error. We completely agree with this comment. However, it should be noted that large-area film formation of PQDs or traditional QDs is rarely reported. The film morphology is highly affected by the fluid dynamics of the solvents. It is still very challenging to manipulate solute redistribution during solvent evaporation (*Sci.Adv.*2020, 6, eaba5029; *Nat. Commun.* 2021, 12, 4381). The surface tension, viscosity, and evaporation of the solvent must be controlled to obtain uniform film formation. Our tailored binary-solvent strategy successfully controls the redistribution of QDs by tuning the fluidic properties and drying dynamics, which is essentially different from previous works on modifying the substrate surface energy. In this regard, our work is still very important for the PeLED field.

In the revised manuscript, we modified our statements to make them more accurate. The new abstract reads as follows: *"One of the most attractive properties of metal halide perovskites (MHPs) is their tunable optoelectronic properties and bandgap via composition engineering. Presynthesized perovskite quantum dots (PQDs) are very promising for making films with different compositions, as they decouple perovskite crystallization and film formation processes. However, fabricating large-area uniform films using PQDs is still very challenging due to the complex fluidic dynamics of the solvents. Here, we report a robust film formation approach using an environmentally friendly binary-solvent strategy....."*

Comment 3: The authors do not provide a satisfactory explanation for why the QD film roughness decreases with the optimal co-solvent ratio and once again increases with excess co-solvent. Beyond cartoon schematics, they are encouraged to model the growth behavior and validate this.

Response: We thank the reviewer very much for the suggestions. In the revised manuscript, we provided a clearer explanation and modified our schematic to make it easier to understand. We reinterpret it here as follows:

For the single solvent system, spreading is coupled with drying after the initial capillary-driven spreading. The solvent evaporation rate increases from the apex to the contact line, resulting in a thermal gradient along the surface. In addition, the heat supply through the liquid phase is not uniform due to either the nonuniform thickness of the droplet or wet film. This creates a temperature gradient in the radial direction and causes thermal Marangoni flow. The direction of the thermal Marangoni flow can be outward or inward, depending on the ratio of

thermal conductivities between the liquid and substrate. For pure n-octane and n-hexane, the direction of the thermal Marangoni flow is inward and enhances solvent contraction. Therefore, most PQDs in the droplet accumulate at the center. This phenomenon occurs when the contact line of the droplet is mobile and the PQDs have enough time to migrate with the solvent (Fig. 3-6a).

For the binary-solvent system (proper n-hexane), the concentration of n-hexane at the edge of the ink droplets is lower than that at the center because of the faster solvent evaporation edge and the higher volatility of n-hexane. Therefore, surface tension gradients arising from spatial variations in n-hexane concentration cause an outward Marangoni flow. The binary solvent with a proper ratio of n-hexane results in spreading of the solvent on the substrate and leads to a uniform film formation process (Fig. 3-6b).

For the binary-solvent system (excess n-hexane), the strong outward Marangoni flow brings solute to the edge of the droplet, leading to thicker perovskite films at the edge (similar to the coffee ring). Moreover, the droplets dry in advance during the spreading process due to the extremely fast evaporation rate of excess n-hexane. This leads to an uncontrollable film-forming process and rough surface morphology (Fig. R3-6c).

Fig. R3-6 | Mechanism of manipulating fluid dynamics using a binary-solvent system. a-c, Schematic illustration of solution droplet evaporation and final film patterns with zero (a), proper (b), and excess (c) ratios of n-hexane. The black arrows represent the flow pattern, and the red arrows represent the moving direction of the unpinned contact line. For the single solvents, inward flow driven by the thermal Marangoni effect induces droplet contraction. For binary

solvents with proper n-hexane, outward flow driven by the concentration Marangoni effect enhances droplet spreading. For the binary solvents with excess n-hexane, the outward flow becomes stronger, but the droplet dries in advance during the spreading process due to the extremely fast evaporation rate of excess n-hexane. **d-f**, Simulation of the velocity vector of solvent flow when the droplets reach equilibrium contact angles with n-hexane ratios of 0 vol% (d), 20 vol% (e), and 80 vol% (f). The blue arrows represent the velocity vector, and the red circle represents the edge position. **(We replaced Fig. 1 in the original manuscript with the above figure)**

We also followed the reviewer’s suggestion and simulated the solvent flow in the droplets in collaboration with Ting Si, a well-known expert in the field of fluid mechanics at our university. The simulation uses the COMOSL platform based on the finite element method. The computational domain is shown in Fig. R3-7. Both the top view (Fig. R3-6d-f) and the side view (Fig. R3-8) results confirm the existence of outward Marangoni flow using a binary solvent and the overstrong outward Marangoni flow with excess n-hexane.

Fig. R3-7 | Computation configuration of the droplets. **(We added this figure as Supplementary Fig. 2 in the revised manuscript)**

Fig. R3-8 | Simulation (side view) of velocity vector distribution when the droplets reach equilibrium contact angle with an n-hexane ratio of 0 vol% (a), 20 vol% (b), and 80 vol% (c).

The blue arrows represent the velocity vector, and the red circle represents the interface position. **(We added this figure as Supplementary Fig. 3 in the revised manuscript)**

We modified the manuscript accordingly:

1. We added the simulation data as Fig. 1d-f in the revised manuscript and updated the schematics of the film-forming process. We added the simulation details as Supplementary text 1 in the revised Supplementary information.
2. On page 5, line 14 of the revised manuscript, we added the following: *“We simulated the solvent flow in the droplets using the COMOSL platform based on the finite element method (Supplementary text 1). The computational domain is shown in Supplementary Fig. 2. Both the top view (Fig. 1 d-f) and the side view (Supplementary Fig. 3) results confirm the shift from inward to outward Marangoni flow after mixing n-octane with n-hexane (Fig. 1d,e), as well as the overstrong outward Marangoni flow and weakened droplet spreading with excess n-hexane (Fig. 1f).”*
3. We rewrote Section 1 of the revised manuscript *“Mechanisms of manipulating fluidic properties of perovskite inks”* to make it clearer.

On page 4, line 13 of the revised manuscript, we added: *“For a single solvent system (Fig. 1a), such as n-hexane or n-octane, preferential solvent evaporation results in an inward thermal gradient from the apex to the edge of the droplets because the thickness of the droplet is nonuniform and the thermal conductivity of the substrate is generally much larger than these short-chain alkane solvents³⁶. This inward temperature gradient causes a surface tension gradient and thus an inward Marangoni flow.”*

On page 5, line 1 of the revised manuscript, we added: *“To suppress the inward Marangoni flow induced by the temperature gradient, we intentionally blend n-hexane with higher volatility and lower surface tension than n-octane (Fig. 1b). Therefore, the n-hexane concentration-caused outward Marangoni flow plays a leading role and results in solvent spreading and uniform solute redistribution.”*

On page 5, line 11 of the revised manuscript, we added: *“For the binary-solvent with excess n-hexane (Fig. 1c), the overstrong outward Marangoni flow brings solute to the edge of the droplet and induces a thicker film at the edge. Moreover, the spreading process of the droplets is strongly coupled with the fast drying step due to the extremely fast evaporation rate of n-hexane, and the droplet dries during the spreading stage. This leads to weakened droplet spreading and an uncontrollable solute redistribution process.*

We simulated the solvent flow in the droplets using the COMOSL platform based on the finite element method (Supplementary text 1). The computational domain is shown in Supplementary Fig. 2. Both the top view (Fig. 1 d-f) and the side view (Supplementary Fig. 3) results confirm the shift from inward to outward Marangoni flow after mixing n-octane with n-hexane (Fig. 1d,e), as well as the overstrong outward Marangoni flow and weakened droplet spreading with excess n-hexane (Fig. 1f).”

Reviewer #4:

Comments: This work reports a solvent engineering to deposit large-area and uniform perovskite quantum dots films of different compositions by blade coating. A binary-solvent strategy is developed which allows to achieve low surface energy, thereby enhancing its wettability, and eventually improves the film homogeneity. The authors carefully organized the manuscript and tried to sell a sound story. However, the entire work focuses too much on the engineering aspects with little scientific contribution in the field. In addition, the EQEs of the prepared small-area LED devices are rather decent as compared to the state-of-the-art perovskite QLED devices. As such, I would recommend this manuscript to be considered in a more specific journal based on, additionally, the following comments.

Response: We thank the reviewer for the criticism. **The strong spread ability of the binary solvent used in our work is not related to the surface energy and wettability of the substrate.** As shown in the following figure, there are two essentially different approaches to improve the spreading of solvents on a substrate (*Phys. Rev. Lett.* 2021, 127, 024502). As the reviewer mentioned, one approach is to improve the wetting of the substrate by reducing its surface energy (Fig. R4-1a). **The second approach is to manipulate the fluidic and evaporation dynamics of the solvent (Fig. R4-1b), which is the case in our work (see our detailed response to Comment #2).** We tested four substrates with different surface energies: glass, PEDOT:PSS, poly(9-vinylcarbazole) (PVK), and poly-TPD. As shown in Fig. R4-2, the influence of the substrate is very minor because of the high spreading ability of our binary solvents. We obtained almost identical deposition patterns on these different surfaces.

Fig. R4-1 | Schematic diagram of (a) direct evaporation (capillary)-driven spreading and (b) Marangoni-driven spreading. (Figure is adapted from *Phys. Rev. Lett.* 2021, 127, 024502)

Fig. R4-2 | Morphology of PQD (20 vol% n-hexane) films on different substrates. The volume of PQD inks is 3 μl , and the scale bar is 1 cm. The insets are photographs of the contact angle between water and different substrates. **(We added this figure as Supplementary Fig. 12 in the revised manuscript)**

To make the differences between our approach and previous approaches tuning surface energy clearer, we redrew the schematics in Fig. 1 of the revised manuscript as shown below. We also followed reviewer #3's suggestion and simulated the solvent flow in the droplets using COMOSL to provide an in-depth study on the mechanism of manipulating the solvent flow. The computational domain is shown in Fig. R3-7. Both the top view (Fig. R3-6d-f) and side view (Fig. R3-8) results confirm the existence of outward Marangoni flow using a binary solvent with a proper ratio of n-hexane. The simulation results also show overstrong outward Marangoni flow with excess n-hexane, which results in rough films.

In short, we provide a new solvent system to manipulate the fluidic properties of PQD inks for the first time. We believe that the solvent design, working mechanism, and demonstration of large-area WPeLEDs could contribute significantly to the PeLED field. Regarding device performance, to the best of our knowledge, there is no report on blade-coated red PeLEDs and white PeLEDs. We think that the peak EQE of 15.3% for red PeLEDs and 10.6% for white PeLEDs are very decent values.

Fig. R4-3 | Mechanism of manipulating fluid dynamics using a binary-solvent system. a-c, Schematic illustration of solution droplet evaporation and final film patterns with zero (a), proper (b), and excess (c) ratios of n-hexane. The black arrows in the side view and top view represent the flow pattern, and the red arrows represent the moving direction of the unpinned contact line. For the single solvents, inward flow driven by the thermal Marangoni effect induces droplet contraction. For binary solvents with proper n-hexane, outward flow driven by the concentration

Marangoni effect enhances droplet spreading. For the binary solvents with excess n-hexane, the outward flow becomes stronger, but the droplet dries in advance during the spreading process due to the extremely fast evaporation rate of excess n-hexane. d-f, Simulation of the solvent velocity vector when the droplets reach equilibrium contact angles with n-hexane ratios of 0 vol% (d), 20 vol% (e), and 80 vol% (f). The blue arrows represent the velocity vector, and the red circle represents the edge position. **(We replaced Fig. 1 in the original manuscript with the above figure)**

Fig. R4-4 | Computation configuration of the droplets. **(We added this figure as Supplementary Fig. 1 in the revised manuscript)**

Fig. R4-5 | Simulation (side view) of velocity vector distribution when the droplets reach equilibrium contact angle with an n-hexane ratio of 0 vol% (a), 20 vol% (b), and 80 vol% (c). The blue arrows represent the velocity vector, and the red circle represents the interface position. **(We added this figure as Supplementary Fig. 3 in the revised manuscript)**

Comment 1: In the abstract, the authors highlight “Presynthesized perovskite quantum dots (PQDs) are used to decouple the perovskite crystallization and film formation processes”. This

sentence is obviously meaningless, as we know that film formation of quantum dots from a dispersion solution doesn't involve any nucleation and crystallization processes. The authors don't necessarily relate the deposition of quantum dots film to the perovskite formation from precursor-based solutions.

Response: We thank the reviewer very much for pointing out the misunderstanding statements, and we apologize for this error. We completely agree with this comment. However, it should be noted that large-area film formation of PQDs or traditional QDs is rarely reported. The film morphology highly depends on the fluid dynamics of the solvents. It is still very challenging to manipulate solute redistribution during solvent evaporation (*Sci.Adv.*2020, 6, eaba5029; *Nat. Commun.* 2021, 12, 4381). The surface tension, viscosity, and evaporation of the solvent must be controlled to obtain uniform film formation. Our tailored binary-solvent strategy successfully controls the redistribution of QDs by tuning the fluidic properties and drying dynamics, which is essentially different from previous works on modifying the substrate surface energy. In this regard, our work is still very important for the PeLED field.

In the revised manuscript, we modified our statements to make them more accurate. The new abstract reads as follows: *“One of the most attractive properties of metal halide perovskites (MHPs) is their tunable optoelectronic properties and bandgap via composition engineering. Presynthesized perovskite quantum dots (PQDs) are very promising for making films with different compositions, as they decouple perovskite crystallization and film formation processes. However, fabricating large-area uniform films using PQDs is still very challenging due to the complex fluidic dynamics of the solvents. Here, we report a robust film formation approach using an environmentally friendly binary-solvent strategy.....”*

Comment 2: Solvent engineering, including binary and ternary solvent strategies, has been well studied in spin-coating to deposit large-area perovskite LED devices. The principle behind the solvent engineering for blade-coating in the present work should be the same as previous studies on spin-coating, which is to reduce surface energy and increase wettability of the formulation.

Response: We thank the reviewer for this comment. As we mentioned above, there are two approaches to enhance the spreading of droplets: capillary flow (depending on the surface energy) and Marangoni flow (depending on the fluidic properties of the solvents). Most previous works focused on substrate modification to increase the capillary flow by adjusting the surface energy and ultimately improve wettability. **In direct contrast, our strategy is to induce an outward Marangoni flow using the binary-solvent system, which is not related to the wetting of the substrate.** We can deposit uniform films on nonwetting surfaces (Fig. R4-2).

Marangoni flow can be classified as inward Marangoni flow and outward Marangoni flow when using a binary solvent, as shown in Fig. R4-6 (*Physics Reports* 2022, 960, 1-37). The following are the mechanisms:

For outward Marangoni flow: As shown in Fig. R4-6a, the Marangoni effect resulting from the surface tension gradient toward the edge can enhance the advancing motion of the contact line (*Physics Reports* 2022, 960, 1-37), *i.e.*, binary solvents spread faster than single solvents (*Langmuir* 1998, 14, 9, 2554-2561). Furthermore, complete spreading also occurs in binary solvents even if each solvent does not completely spread alone. **The concentration Marangoni effect induced by preferential evaporation enables the droplet to overcome the energy barrier and realize complete wetting, such as our binary-solvent n-octane/n-hexane and the binary-solvent phenetole/chloroform (*Langmuir* 1987, 3, 4, 519-524).**

For the inward Marangoni flow: As shown in Fig. R4-6b, when the less volatile solvent has lower surface tension, the surface tension near the contact line becomes smaller than that near the droplet apex. As a result, the surface tension gradient toward the apex induces Marangoni flow in the same direction. The inward Marangoni effect drags back the contact line and contracts the droplet. In this case, droplet spreading is retarded, and the spreading rate is lower than that of both single solvents (*Langmuir* 1987, 3, 4, 519-524). Even on high surface energy substrates, the binary solvent may exhibit an apparent nonzero contact angle (*Nature* 2015, 519, 446-450).

Fig. R4-6 | (a) Spreading or (b) contraction of a binary-solvent droplet induced by the concentration Marangoni effect due to preferential evaporation of the low surface tension solvent or the high surface tension solvent at the contact line, respectively. Positive (negative) gamma (γ) indicates high (low) surface tension. (Figures are adapted from *Physics Reports* 2022, 960, 1–37)

Different from spin coating, blade coating has a very different film-forming mechanism. We believe our work represents a significant step forward for blade coating. Antisolvent is usually used in the spin coating process to precipitate perovskites. Blade coating usually uses other assistant techniques, such as N₂ knife, vacuum, and substrate heating, to accelerate solvent evaporation. These additional processes are all very difficult to control when increasing the device area. Our work is the first report that does not need any assistant treatment for the film formation process, which is much easier to scale up.

We modified the manuscript accordingly:

1. On page 6, line 13 of the revised manuscript, we added the critical differences between our approach and previous approaches on cosolvent and modified the surface energy: “It is noted that the binary-solvent approach has been adopted in preparing perovskite films or crystals³⁸⁻⁴⁰. The purpose of using a binary solvent includes tuning the solvent-solute interaction strength³⁸, tuning the evaporation speed³⁹, tuning the nucleation and crystallization process of perovskites⁴⁰, etc. Our binary solvent is designed to manipulate the fluidic properties of the solvent to achieve uniform large-scale PQD deposition,

which is essentially different. The droplet spreading using our binary-solvent system is also essentially different from the works modifying the surface energy of the substrate⁴⁰. As shown in Supplementary Fig. 12, we can obtain almost the same uniform films on substrates with very different surface energies, which demonstrates that the binary-solvent strategy is a robust approach.”

Comment 3: Related works should be cited in the design of the white LED in combination with sky blue perovskite LED devices.

Response: We thank the reviewer for the valuable suggestions and apologize for not citing related important works. As far as we know, there is only one study about this structure design in the PeLED field from Prof. Hin-Lap Yip & Ziming Chen’s group (*Joule* 2021, 5, 456-466). We cited it as ref. # 16 on page 2, line 11, and page 10, line 12 of the revised manuscript.

Reviewer #5:

Comments: This manuscript as noted by Shi and co-workers demonstrated a binary-solvent strategy (n-octane and n-hexane) for fabrication of large-area perovskite light-emitting diodes (PeLEDs). With this binary solvent (n-octane and n-hexane) recipe, a blade-coated uniform perovskite quantum dot (PQD) thin film forming process is obtained. As a result, a good external quantum efficiency (EQE) of 15.3% is achieved in PeLEDs with the blade-coated PQD thin film. Large-area white PeLEDs (WPeLEDs) with area of 28 cm² by coupling a sky-blue PeLED with a layer of red PQDs as a downconverter is further demonstrated, and an EQE of 10.6% is realized. However, similar strategy has already been reported by the other groups (*Advanced Materials*, 2022, adma.202107798; *Micromachines*, 2022, 13, 983; *Nano Research*, 2021, 14, 4125–4131), and large-area PeLEDs have already been presented by the same group (*Advanced Materials*, 2022, adma.202108939; *Nature Communications*, 2021, 12, 147). This paper mainly focuses on optimizing the fabrication processes of the devices. Although the performance of PeLEDs is improved, the mechanisms are not so clear and further verifications are needed. Considering the limited novelty (fabrication processes optimization) of the paper and very little scientific contribution to field, I do not recommend its to be published in high impact journals like *Nature Communications*. The paper seems to be more suited for other specialized journals after addressing below issues, eg. *Light: Science & Applications*.

Response: We thank the reviewer for the criticism. We understand the reviewer's concern and want to respond to this comment from the following four aspects:

1. **Our strategy is essentially different from that of previous works.** Those previous works focused on inkjet printing patterned perovskites, and their solvent system was designed to not spread the perovskite inks. Their solvent strategies aimed to address the challenge of coffee rings, which exist in their solvent systems because the contact line of their solvent is pinned. **In direct contrast**, we aim to make large-area PeLEDs for flat panel lighting, and therefore, we designed our solvent system to spread the perovskite inks. In our case, the solute transports to the center of the droplet driven by the inward Marangoni flow, which is actually the opposite of the coffee ring pattern. Therefore, it is unfair to compare our strategy with those strategies because the motivations and mechanisms between them are opposite.
2. **This work is a breakthrough in the PeLED field and is essentially different from our previous works** (*Advanced Materials*, 2022, adma.202108939; *Nature Communications*, 2021, 12, 147). In our previous works, we demonstrated that the complex and diverse film formation process of perovskites with different compositions often results in poorly controlled film morphology and a high density of defects, especially in large-area films made by mass fabrication techniques. Separately optimizing MHP films with different compositions consumes much additional effort and cost, which is unfavorable for the commercialization of perovskite devices. In this work, we solved this critical issue by using PQDs and developed a binary-solvent system to prepare high-quality perovskite films with different compositions using exactly the same fabrication parameters.
3. **Our work is not only about engineering but also provides in-depth scientific contributions about the fluidic and evaporation dynamics of PQD inks, which have not been reported before.** We reveal the fluidic and evaporation dynamics using green solvents (halogen-free, nonbenzene and very low toxicity) and the reason why the films are not uniform using a single solvent. We solve this issue by inventing a new binary-solvent system that is essentially different from previous works. To the best of our

knowledge, our binary-solvent strategy is the first report to fabricate large-area perovskite thin films with different compositions using the same fabrication parameters and will contribute significantly to the PeLED field.

To further provide in-depth insight into the mechanism of the film formation process, we performed a simulation of the solvent flow in the droplets using COMOSL (Fig. R5-1). The results confirm the existence of outward Marangoni flow using a binary solvent and the overstrength outward Marangoni flow with excess n-hexane (see our response to Comment #1).

- 4. The first demonstration of large-area WPeLEDs is a milestone in the field of PeLEDs. This represents a significant step toward SSL using perovskites as emitters.** The fabrication of large-area white LEDs using a solution process for flat panel lighting has been pursued in both OLEDs and QLEDs for decades. However, there has been limited success to date due to the uncontrollable film quality prepared by the solution process. Although WPeLEDs are promising for next-generation lighting, their future remains unclear due to the lack of demonstration for large-scale WPeLED lighting panels. We overcome the critical issue of uncontrollable fluidic and drying properties of perovskite inks and obtain efficient and bright large-area WPeLEDs. Our work provides strong evidence for the feasibility of using perovskites as emitters. Moreover, our WPeLEDs use sky-blue PeLEDs (491 nm) with low ophthalmic toxicity as exciting light sources, which are much healthier than GaN-based light sources.

With the above achievements, we believe our work will attract broad research interests and have a high impact on thin film LEDs and the perovskite field. We trust that *Nature Communications* is the best avenue to publish our work and kindly request your consideration of our manuscript.

Comment 1: In Figure S3., the authors claims that 0 vol% and 100 vol% is capillary spreading, 20 vol% and 40 vol% is the Marangoni enhanced spreading range, and the 60 vol% and 80 vol% is evaporation weakened spreading. However, if there is a fluid volatilization, there is a Marangoni flow effect. The detailed mechanism and why the PQDs (different types) could form uniform thin films, how is the PQDs packing during the processing are not been clearly investigated and understood.

Response: We thank the reviewer for the comments. We respond to this comment from the following two aspects:

1. The mechanism of the film-forming process using a binary solvent.

It is true that there is a Marangoni flow effect if there is fluid volatilization. **However, the Marangoni flow direction is different for different solvent systems, which results in very different film morphologies.**

For the single solvent system, spreading is coupled with drying after the initial capillary-driven spreading. The solvent evaporation rate increases from the apex to the contact line, resulting in a thermal gradient along the surface. In addition, the heat supply through the liquid phase is not uniform due to either the nonuniform thickness of the droplet or wet film. This creates a temperature gradient in the radial direction and causes thermal Marangoni flow. The direction of the thermal Marangoni flow can be outward or inward, depending on the ratio of thermal conductivities between the liquid and substrate. For pure n-octane and n-hexane, the direction of the thermal Marangoni flow is inward and enhances solvent contraction. Therefore,

most PQDs in the droplet accumulate at the center. This phenomenon occurs when the contact line of the droplet is mobile and the PQDs have enough time to migrate with the solvent (Fig. 5-1a).

For the binary-solvent system (proper n-hexane), the concentration of n-hexane at the edge of the ink droplets is lower than that at the center because of the faster solvent evaporation edge and the higher volatility of n-hexane. Therefore, surface tension gradients arising from spatial variations in n-hexane concentration cause an outward Marangoni flow. The binary solvent with a proper ratio of n-hexane results in spreading of the solvent on the substrate and leads to a uniform film formation process (Fig. 5-1b).

For the binary-solvent system (excess n-hexane), the strong outward Marangoni flow brings solute to the edge of the droplet, leading to thicker perovskite films at the edge (similar to the coffee ring). Moreover, the droplets dry in advance during the spreading process due to the extremely fast evaporation rate of excess n-hexane. This leads to an uncontrollable film-forming process and rough surface morphology (Fig. R5-1c).

Fig. R5-1 | Mechanism of manipulating fluid dynamics using a binary-solvent system. a-c, Schematic illustration of solution droplet evaporation and final film patterns with zero (a), proper (b), and excess (c) ratios of n-hexane. The black arrows represent the flow pattern, and the red arrows represent the moving direction of the unpinned contact line. For the single solvents, inward flow driven by the thermal Marangoni effect induces droplet contraction. For binary solvents with proper n-hexane, outward flow driven by the concentration Marangoni effect

enhances droplet spreading. For the binary solvents with excess n-hexane, the outward flow becomes stronger, but the droplet dries in advance during the spreading process due to the extremely fast evaporation rate of excess n-hexane. **d-f**, Simulation of the velocity vector of solvent flow when the droplets reach equilibrium contact angles with n-hexane ratios of 0 vol% (d), 20 vol% (e), and 80 vol% (f). The blue arrows represent the velocity vector, and the red circle represents the edge position. **(We replaced Fig. 1 in the original manuscript with the above figure)**

We also followed reviewer #3's suggestion and simulated the solvent flow in the droplets in collaboration with Ting Si, a well-known expert in the field of fluid mechanics at our university. The simulation uses the COMOSL platform based on the finite element method. The computational domain is shown in Fig. R5-2. Both the top view (Fig. R5-1d-f) and the side view (Fig. R5-3) results confirm the existence of outward Marangoni flow using a binary solvent and the overstrong outward Marangoni flow with excess n-hexane.

Fig. R5-2 | Computation configuration of the droplets. **(We added this figure as Supplementary Fig. 2 in the revised manuscript)**

Fig. R5-3 | Simulation (side view) of velocity vector distribution when the droplets reach equilibrium contact angle with an n-hexane ratio of 0 vol% (a), 20 vol% (b), and 80 vol% (c).

The blue arrows represent the velocity vector, and the red circle represents the interface position. (We added this figure as Supplementary Fig. 3 in the revised manuscript)

2. PQD packing in the perovskite films.

We studied the surface morphology of blade-coated PQD films with different n-hexane concentrations. Although the surface roughness varies with the n-hexane ratio, there are no cracks or pinholes on the film surfaces (Fig. S10 c,d). We also measured the surface and cross-section morphology of drop-casted PQD films (Fig. R5-4). The morphologies are consistent with blade-coated films. Compared to binary solvents, PQD films made using single solvents are significantly rougher.

In fact, in meniscus coating processes, such as blade-coating and slot-die coating, meniscus-guided assembly has an inherent advantage of capillary attraction, which gives rise to the self-draining effect. As illustrated in Fig. R5-5, compared with spin coating, the self-draining behavior effectively extrudes solvent residuals in CQD films, which results in compact packing of CQDs and minimizes the formation of cracks and pinholes during the drying process of wet films.

Fig. S10 c,d | AFM (c) and SEM (d) images of blade-coated PQD films with different n-hexane concentrations. Scale bars are 5 μm and 500 nm, respectively.

Fig. R5-4 | Cross-section (top panel) and surface SEM images (bottom panel) of drop-casted PQD films with different concentrations of n-hexane. Scale bars are 200 nm. (We added this figure as Supplementary Fig. 10 in the revised manuscript)

Fig. R5-5 | Illustration of the meniscus-guided coating and spin-coating process for the preparation of CQD solids. (Adapted from *Nat. Commun.* 2021, 12, 4381)

We modified the manuscript accordingly:

1. On page 7, line 9 of the revised manuscript, we added: “The PQDs are packed densely in all films, although the roughness varies with the n-hexane ratio (Supplementary Fig. 10).”
2. We rewrote Section 1 of the revised manuscript “Mechanisms of manipulating fluidic properties of perovskite inks” to make it clearer.

On page 4, line 13 of the revised manuscript, we added: “For a single solvent system (Fig. 1a), such as n-hexane or n-octane, preferential solvent evaporation results in an inward thermal gradient from the apex to the edge of the droplets because the thickness of the droplet is nonuniform and the thermal conductivity of the substrate is

generally much larger than these short-chain alkane solvents³⁶. This inward temperature gradient causes a surface tension gradient and thus an inward Marangoni flow.”

On page 5, line 1 of the revised manuscript, we added: “To suppress the inward Marangoni flow induced by the temperature gradient, we intentionally blend n-hexane with higher volatility and lower surface tension than n-octane (Fig. 1b). Therefore, the n-hexane concentration-caused outward Marangoni flow plays a leading role and results in solvent spreading and uniform solute redistribution.”

On page 5, line 11 of the revised manuscript, we added: “For the binary-solvent with excess n-hexane (Fig. 1c), the overstrong outward Marangoni flow brings solute to the edge of the droplet and induces a thicker film at the edge. Moreover, the spreading process of the droplets is strongly coupled with the fast drying step due to the extremely fast evaporation rate of n-hexane, and the droplet dries during the spreading stage. This leads to weakened droplet spreading and an uncontrollable solute redistribution process.

We simulated the solvent flow in the droplets using the COMOSL platform based on the finite element method (Supplementary text 1). The computational domain is shown in Supplementary Fig. 2. Both the top view (Fig. 1 d-f) and the side view (Supplementary Fig. 3) results confirm the shift from inward to outward Marangoni flow after mixing n-octane with n-hexane (Fig. 1d,e), as well as the overstrong outward Marangoni flow and weakened droplet spreading with excess n-hexane (Fig. 1f).”

Comment 2: What about the long-term storage stability of the n-octane and n-hexane-based PQD inks?

Response: We apologize for missing these data. We tested the stability of PQD inks with different n-hexane concentrations for one month, and these PQD inks still maintained good performance due to the good ligand-mediated solvation of PQDs in our binary solvents (Fig. R5-6).

Fig. R5-6 | Photographs of FAPbI₃, CsPbI₃, and CsPbBr₃ QDs dispersed in a single solvent system or binary-solvent system with different n-hexane concentrations under room light (left) and a UV lamp at 365 nm (right) after storage in air at 4 °C for 30 days. The concentration of PQDs was 30 mg/mL. **(We added this figure as Supplementary Fig. 6 in the revised manuscript)**

We modified the manuscript accordingly:

1. On page 6, line 4 of the revised manuscript, we added: “...and the PQD inks maintain good performance after storage for 30 days due to the good ligand-mediated solvation of PQDs in our binary solvents (Supplementary Fig. 6)”

Comment 3: The packing, defects and electrical properties of the PQD thin film have not been carefully investigated.

Response: We thank the reviewer for the suggestion. The packing of the QDs has been illustrated in our response to Comment #1.

To further investigate the optoelectrical properties of the PQD films, we measured the steady-state and transient PL of the PQD films with different n-hexane concentrations. The PQD films with optimized n-hexane concentrations (20 vol% and 40 vol%) showed a higher PLQY value and extended PL lifetime compared with the PQD films with single solvents. These differences should arise from the poor film morphology and surface defects. We also calculated the radiative recombination rate (k_{rad}) and the nonradiative recombination rate (k_{nonrad}) of each sample using the following equations: $1/\tau_{\text{avg}} = k_{\text{rad}} + k_{\text{nonrad}}$ and $\text{PLQY} = k_{\text{rad}}/(k_{\text{rad}} + k_{\text{nonrad}})$. As shown in Table R5-1, PQD films with binary solvents showed decreased k_{nonrad} compared with films with single solvents, and the k_{rad} shows a significant correlation with the film morphology. These results imply that the recipe of PQD inks affects the morphology and recombination dynamics, and the PQD films fabricated with our optimized binary-solvent system realized a uniform film formation process and relatively fewer defects.

Fig. R5-7 | (a) Steady-state PL spectra and (b) transient PL decay curves of PQD films with different n-hexane concentrations. The intensity of the steady-state PL spectra is normalized according to their PLQY values. **(We added this figure as Supplementary Fig. 11 in the revised manuscript)**

Table R5-1 | Summarized average PL lifetime (τ_{avg}), PLQY, radiative recombination rate k_{rad} and nonradiative recombination rate k_{nonrad} of PQD films with different n-hexane concentrations. **(We added this table as Supplementary Table 1 in the revised manuscript)**

	τ_{avg} [ns]	PLQY[%]	k_{rad} [s^{-1}]	k_{nonrad} [s^{-1}]
0 vol%	34.4	34.6	1.0×10^7	1.9×10^7
20 vol%	43.5	46.8	1.1×10^7	1.2×10^7
40 vol%	41.8	44.6	1.1×10^7	1.3×10^7
60 vol%	40.7	42.4	1.0×10^7	1.4×10^7
80 vol%	38.5	41.9	1.1×10^7	1.5×10^7
100 vol%	34.9	35.1	1.0×10^7	1.9×10^7

For the electrical properties, the current density (J) and luminance (L) curves as a function of voltage (V) for CsPbI₃ devices are shown in Fig. 4d. The device with 20 vol% hexane has the lowest turn-on voltage of 2.3 V at a luminance of 1 cd/m². Additionally, the peak EQE with 20 vol% hexane reaches 15.3%, much higher than that of the control device of 8.0%, due to the better film uniformity and optoelectronic properties (Fig. 4f). We also demonstrated ultralarge PeLEDs with a device area of 4×7 cm². Under both low and high driving voltages, the large-area PeLEDs show remarkably uniform light emission across the entire 28 cm² device area, demonstrating the excellent optoelectronic property uniformity of the large-area blade-coated films.

We modified the manuscript accordingly:

1. On page 7, line 9 of the revised manuscript, we added the following: “*The PQD films made from 20 vol% and 40 vol% n-hexane also have better optoelectronic properties, such as longer carrier lifetimes and higher PLQYs (Supplementary Fig. 11 and Supplementary Table 1).*”

Comment 4: More detail of PQD ink preparation and PeLED device fabrication should be further supplied to make this work more repeatable to the readers.

Response: We thank the reviewer for this suggestion. We have tried to add all experimental details that we can think of in the revised version.

1. We added the synthesis and purification process of CsPbBr₃ and FAPbI₃ QDs:

“For CsPbBr₃ QDs, the Cs-OA precursor was prepared similarly to the CsPbI₃ QDs mentioned above. In a reaction flask, 700 mg of PbBr₂ and 1,400 mg of ZnBr₂ were mixed with 14 ml of OAm/OA mixture and 40 ml of ODE, followed by vacuum drying at 100 °C for 30 min. The lead halide precursors were kept under a nitrogen atmosphere at 210 °C until all solids were dissolved. Then, 11.0-11.8 ml of Cs-OA precursor solution was swiftly injected into the flask containing the lead halide precursor. After 10 s, the solution was cooled using an ice water bath. The solution was centrifuged at 7,800 r.p.m. to remove unreacted precursors. For the purification process, the precipitant was collected and dispersed in 7 ml of n-hexane followed by the addition of 20 ml of methyl acetate. The mixture was centrifuged at 7,800 r.p.m. to collect the precipitants and dissolved in 7 ml n-hexane. This procedure was repeated once or twice. Finally, the supernatant was stored at 4 °C before use.

For the FAPbI₃ QDs, 521 mg FA-acetate and 10 ml OA were placed in a flask and degassed by heating at 120 °C for 1 h and then stirred at 80 °C for 1 h under N₂ to prepare the

FA-OA precursor. In another flask, 175 mg PbI₂ was mixed with 1 ml OAm, 1 ml OA and 10 mL ODE followed by degassing at 100 °C for 1 h. After the PbI₂ salt was completely dissolved, the solution was cooled to 80 °C. Subsequently, 1 mL of FA-OA precursor was injected. After 5 s, the reaction flask was cooled in an ice bath. For the purification process, the solution was centrifuged at 12 000 rpm for 10 min. The obtained precipitate was then added to 7 mL n-hexane and 7 mL ethyl acetate and then centrifuged at 12 000 rpm for 15 min. The precipitate was collected and dispersed in 2 mL n-hexane and centrifuged at 12 000 rpm for 15 min. Finally, the supernatant was stored at 4 °C before use.”

2. We updated the blade-coating details of the LED fabrication and characterization:

“For red PeLEDs, the PEDOT:PSS solution was spun or blade-coated onto the surface of the cleaned ITO substrate and baked at 140 °C for 20 min under ambient conditions. The poly-TPD, 6 mg/ml in chlorobenzene, was spin-coated (2000 rpm) or blade-coated on the PEDOT:PSS layer, followed by annealing at 150 °C for 20 min. The poly-TPD layer was treated with O₂ plasma for 6 s to improve wetting. We blade-coated the CsPbI₃ QDs in air with a temperature and humidity of approximately 25 °C and 35%, respectively. The speed of the applicator was set as 50 mm/s. The distance between the applicator and substrate is approximately 2-3 μm. The blade-coated PQD films were dried at 60 °C for 10 min to fully dry the film. The TPBi, LiF, and Al layers were sequentially thermally evaporated on top of the perovskite film with thicknesses of 40, 1.2, and 100 nm, respectively.

For white PeLEDs, the PVK solution (3 mg/mL) was prepared by spin-coating (2000 rpm) or blade-coating and then baked at 120 °C for 20 min. The PVK layer was treated by O₂ plasma for 6 s to improve wetting. Then, the sky-blue perovskite precursor solution was blade-coated onto the PVK films in air and dried by applying an N₂ knife for 1 min at a substrate temperature of 50 °C. The speed of the applicator was set as 50 mm/s. The distance between the applicator and substrate is approximately 2-3 μm. There was no further annealing process to avoid grain growth and reduction of the PL intensity. The TPBi, LiF, and Al layers were sequentially thermally evaporated on top of the perovskite film with thicknesses of 40, 1.2, and 100 nm, respectively. Then, the red PQD inks (A: 180 mg/ml, B: 140 mg/ml, and C: 120 mg/ml) were blade-coated on the bottom of the substrate. The blade-coated parameters are consistent with the emission layer.”

REVIEWER COMMENTS

Reviewer #1 (Remarks to the Author):

The revisions are satisfactory and I am happy to recommend the acceptance of the manuscript for publication.

Reviewer #4 (Remarks to the Author):

The authors have made every effort to revise the manuscript to highlight the scientific parts of the work. The considerable supplementary information now explains more reasonable the choice of binary solvents and the related mechanism behind. On these bases, I would support the publication of the manuscript at this stage. Two other comment: In the abstract, the authors should include one or two more scientific sentences explaining the functionality and mechanism of the binary solvent in producing uniform films, rather than general introduction of the outcome of their method. In the main text, the authors are encouraged to highlight the guidelines / principles of choosing mixed (binary or triple) solvents to deposit uniform QD-LED films, other than the specific two used in this work.

Reviewer #5 (Remarks to the Author):

Although the authors made some improvements for the manuscript with the help from the referees' comments and suggestions, a revision is still needed. The authors utilize COMOSL to modeling the solvent flow to further reveal the mechanisms, however, the other components in the ink, like the perovskite quantum dots and detached ligands should be also considered. Especially, how is the solvent flow influencing the perovskite quantum dots packing during the blade coating and drying process should be further investigated and explained.

Response Letter

We thank all reviewers for their acknowledgment of our revised work. It is very encouraging for us. We modified the manuscript according to reviewer #4's comments, and performed additional experiments to address reviewer #5's concerns.

Reviewer #1:

Comments: The revisions are satisfactory and I am happy to recommend the acceptance of the manuscript for publication.

Response: We appreciate the reviewer's acknowledgment of our revised work. We also thank the reviewer very much for the technical issues raised before, which have helped us improve the quality of our manuscript.

Reviewer #4:

Comments: The authors have made every effort to revise the manuscript to highlight the scientific parts of the work. The considerable supplementary information now explains more reasonable the choice of binary solvents and the related mechanism behind. On these bases, I would support the publication of the manuscript at this stage. Two other comment: In the abstract, the authors should include one or two more scientific sentences explaining the functionality and mechanism of the binary solvent in producing uniform films, rather than general introduction of the outcome of their method. In the main text, the authors are encouraged to highlight the guidelines / principles of choosing mixed (binary or triple) solvents to deposit uniform QD-LED films, other than the specific two used in this work.

Response: We thank the reviewer for the time and appreciate the reviewer's acknowledgment of our revised work.

We followed the suggestion and modified the manuscript accordingly:

1. We added the following sentence in abstract to explain the mechanism of forming uniform films: *“Nonbenzene solvents, n-octane and n-hexane, are mixed to manipulate the fluidic and evaporation dynamics of the PQD inks, resulting in balanced Marangoni flow, enhanced spreadability of PQD inks, and uniform solute redistribution.”*
2. On page 4, line 5 of the revised manuscript, we added the guidelines/principles of choosing mixed solvents to deposit uniform QD-LEDs: *“ Binary or ternary solvents has advantages of tunable viscosity and surface tension gradient to control the solvent flow of the wet film for uniform solute deposition..... The selection principle of mixed solvents should include the following considerations: 1. Dispersity and stability: the mixed solvents should well mediate with ligands to form a stable ink. 2. Evaporation rate: the mixed solvents should have a suitable boiling point to avoid forming nonuniform film caused by fast solvent evaporation, and also avoid uncontrolled deposition patterns (such as coffee ring) resulting from a slow drying process. 3. Fluidic properties: The solvents should have relatively low viscosity and surface tension to ensure good spreadability for coating uniform thin films. 4. Environmental friendliness: To minimize the environmental and human health impact of coating processes, the solvent used for mixing should be green and low toxicity.”*

Reviewer #5:

Comments: Although the authors made some improvements for the manuscript with the help from the referees' comments and suggestions, a revision is still needed. The authors utilize COMOSL to modeling the solvent flow to further reveal the mechanisms, however, the other components in the ink, like the perovskite quantum dots and detached ligands should be also considered. Especially, how is the solvent flow influencing the perovskite quantum dots packing during the blade coating and drying process should be further investigated and explained.

Response: We thank the reviewer for the acknowledgment of our revised manuscript, and for this valuable comment. We respond to this comment from the following two aspects.

1. COMOSL modeling results using PQD inks.

We followed your suggestion and first measured the fluid properties of PQD inks, such as surface tension and viscosity. We synthesized much more PQDs and for both measurements as the Wilhelmy plate method requires at least 32.4 ml of solution (**Fig. R5-1**). Compared to pure solvents, the PQD inks show negligible increases in surface tension and viscosity (**Fig. R5-2a,b**). We also calculated the vapor-liquid equilibrium phase diagram of PQD inks according to Raoult's Law, the boiling point of PQD inks increases slightly around 2.9-4.2 K compared to binary-solvents (**Fig. R5-2c,d**).

We further simulated the solvent flow using COMSOL based on the physical properties of PQD inks. Both the top view (**Fig. R5-3**) and the side view (**Fig. R5-4**) results show very minor differences in fluid flow behavior between solvents and PQD inks. These results suggest that the influence of PQDs and the possible detached ligands on the fluidic behavior of the solvent is negligible.

Fig. R5-1 | Photographs of CsPbI₃ QDs dispersed in the solvents with different n-hexane concentrations under room light (left) and a UV lamp at 365 nm (right). The volume of each ink is 35 ml. The concentration of PQDs was 30 mg/mL.

Fig. R5-2 | Viscosity and surface tension of binary-solvents (a) and PQD inks (b) (30 mg/ml), vapor-liquid equilibrium phase diagram of binary-solvents (c) and PQD inks (d) (30 mg/ml).

Fig. R5-3 | Simulation (top view) of the velocity vector of solvent flow when the droplets reach equilibrium contact angles. The n-hexane ratios of solvent droplets and PQD ink droplets are 0 vol% (**a, d**), 20 vol% (**b, e**), and 80 vol% (**c, f**). The blue arrows represent the velocity vector, and the red circle represents the edge position. The dash lines in the figures are used for comparison.

Fig. R5-4 | Simulation (side view) of the velocity vector of solvent flow when the droplets reach equilibrium contact angles. The n-hexane ratios of solvent droplets and PQD ink droplets are 0 vol% (**a, d**), 20 vol% (**b, e**), and 80 vol% (**c, f**). The blue arrows represent the velocity vector, and the red circle represents the edge position.

2. How the perovskite quantum dots pack during the blade coating and drying process

Depending on the coating speed and drying speed, the meniscus process can be divided into two regimes, namely the Landau-Levich regime and the evaporation regime. As shown in **Figure R5-5a**, when the n-hexane concentration is low, the coating process is in the Landau-Levich regime, and the film remains wet after coating. The inward Marangoni flow causes the wet film shrink, resulting in a nonuniform center-focused solid film. This center-focused deposited pattern also results from a slow drying process, giving PQDs enough time to migrate with solvent.

For proper n-hexane concentration, the balanced inward and outward Marangoni flows results in uniform solute film (**Figure R5-5b**). Meanwhile, proper n-hexane also accelerates the evaporation rate during the drying stage, and the contact line recedes quickly. Thus PQDs are deposited on the substrate rather than migrate with the receding contact line, resulting in uniform PQDs packing.

The coating process changes to the evaporation regime when there is excess n-hexane, due to the too fast drying speed of the binary solvent. In this case, the PQDs accumulate and form a solid film near the contact line. The over-strong Marangoni flow towards the contact line induces the PQD inks to detach from the receding side of the blader. The over-fast drying also causes most PQDs to pack at the beginning of blading. These problems result in a nonuniform and discontinuous film-formation process (**Figure R5-5c**).

In order to further investigate the effect of solvent flow on the PQD packing during blade-coating and drying processes, we have examined the uniformity of the blade-coated PQD films (**Figure R5-6a,b**). The center-focused packing pattern of the PQDs with 0 vol% n-hexane is confirmed by the thickness profiler and photographs. With the n-hexane concentration increase, the center-focused packing pattern was suppressed and a uniform blade-coating PQD film was obtained with a n-hexane concentration of 20 vol%. However, the blade-coated film becomes

nonuniform with voids and textures when the n-hexane ratio exceeds 30 vol%, in consistent with our analysis above. All thickness profiler, optical images, AFM images, and SEM images confirmed the above trend in packing of PQDs with increasing ratio of n-hexane.

Fig. R5-5 | Schematic illustration of the blade-coating regimes. film-formation process in the Landau-Levich regime when the n-hexane concentration is no or little (a) and proper (b). film-formation process in the evaporation regime when the n-hexane concentration is excess (c). (We added this figure as Supplementary Fig. 13 in the revised manuscript)

Supplementary Fig. 14 | Macroscale and microscale morphology characterizations of blade-coated PQD films. **a**, Photograph of PQD films (5 cm × 5 cm) on a glass substrate under room light (left) and a UV lamp (right). The scale bar is 1 cm. **b**, Thickness profile of PQD films on the glass substrate as shown in (a) from “A” to “B”. **c-e**, Optical microscopy (**c**), AFM (**d**), and SEM (**e**) images of blade-coated PQD films with different n-hexane concentrations. Scale bars are 5 μm and 500 nm, respectively.

We modified the manuscript accordingly:

1. We updated Fig. 1, Fig. S1, and Fig. S3 in the revised manuscript using the new simulation data.
2. We added the optical images as Supplementary Fig. 14c in the revised manuscript.
3. On page 8, line 12 of the revised manuscript, we added: “Depending on the coating speed and drying speed, the meniscus process can be divided into two regimes, namely the Landau-Levich regime and the evaporation regime.”
4. On page 8, line 17 of the revised manuscript, we added: “For the binary-solvent system with proper n-hexane, balanced Marangoni flow results in solvent spreading and uniform solute redistribution as mentioned above (Fig. 3b). Meanwhile, the PQDs are deposited

on the substrate rather than migrate with the receding contact line due to the increased evaporation rates, resulting in uniform PQDs packing (Supplementary Fig. 13b). The coating process changes to the evaporation regime when there is excess n-hexane, due to the too fast drying speed of the binary solvent. In this case, the PQDs accumulate and form a solid film near the contact line. The over-strong Marangoni flow towards the contact line induces the PQD inks to detach from the receding side of the blader. The over-fast drying also causes most PQDs to pack at the beginning of blading. These problems result in a nonuniform and discontinuous film-formation process (Supplementary Fig. 13c). ”